# The dynamics of carbon emissions, energy, income, and life expectancy: Regional comparative analysis

**Frank Osei-Kusi**[1]*, **Cisheng Wu**[1], **Stephen Tetteh**[2], **Wendy Irena Guerra Castillo**[1]

**1** School of Management, Hefei University of Technology, Hefei, Anhui Province, China, **2** School of Applied Economics and Management, University for Development Studies, Tamale, Ghana

* oseifrankhfut@gmail.com

**Data Availability Statement:** All data including the do files are available at github database. https://github.com/kusiman009/Carbon-emission-dynamics.git.

## Abstract

This paper examines the linear effects of economic growth on carbon emissions and their impact on mortality and morbidity rates in specific regions sub-Sahara Africa, Middle-East and North Africa, Europe and Central Asia (SSA, MENA, ECA). By analyzing longitudinal data for 82 panels over 30 years, we investigate the relationships between energy usage, per capita GDP, life expectancy, and carbon emissions. Our estimation results show positive correlations between energy use, carbon production, and life expectancy in both the combined sample and individual regions. However, death rate has a negative relationship with carbon production in the combined sample, MENA, and SSA regions. Per capita GDP positively influences carbon emissions and life expectancy in the combined sample and ECA, MENA, and SSA regions. We also identify asymmetric relationships between per capita GDP and carbon production, with evidence supporting the Environmental Kuznets Curve hypothesis for the combined and ECA samples, and an N-trajectory for SSA. These findings emphasize the importance of region-specific approaches to sustainable development, considering the unique environmental and economic challenges each region faces. Policymakers should consider our research insights when designing policies to mitigate the negative impacts of economic progress on the environment.

## Introduction

Climate change, brought on by the widespread release of carbon emissions, is a serious issue that needs to be tackled right away. According to reports from the World Health Organization (WHO), this phenomena has a wide range of negative implications on public health, including lower life expectancy rates across the globe [1]. The main cause of these large carbon deposits, which are rapidly warming our planet and causing climatic changes, is fossil fuel consumption. Particularly, emissions of dangerous gases, such as greenhouse gases, especially $CO_2$, into our environment are greatly increased by the usage of fossil fuels (i.e., oil, natural gas, and coal). According to the [2], fossil fuel combustion is responsible for an astounding 75% of all greenhouse gas emissions. Moreover, the distribution of carbon emissions, as depicted on the

**Funding:** The authors received no specific funding for this work.

**Competing interests:** The authors have declared that no competing interests exist.

choropleth map, Fig 5 underscores a significant challenge. The MENA (Middle-East and North Africa) region emerges as a notable emitter, while SSA (Sub-Saharan Africa) shows lower emissions. This contrast presents a complex problem of environmental impact and development disparities. Understanding the factors behind these patterns is essential.

Again, income and carbon production have a complicated relationship that differs between different nations and locations. Since people with higher earnings typically have better access to energy-intensive goods and services, like larger homes, more cars, and more frequent air travel, there is a general connection between higher income levels and higher $CO_2$ emissions [3]. However, other studies have also demonstrated that as nations advance and reach high economic levels, they start to shift toward more sustainable and low-carbon modes of growth. The Environmental Kuznets Curve (EKC), which asserts that environmental destruction at first rises as a nation's economy expands but progressively starts to decline as a nation gets wealthy, is based on this basic tenet [4]. The linkage between a country's economic expansion and carbon production has been the subject of numerous research, some of which have discovered proof for the Environmental Kuznets Curve (EKC) theory. For example, [5] found that the EKC theory was accurate for European nations, while [6] supported the EKC hypothesis for residential energy use in developing nations. However, other researchers have found evidence of an inverted N-trajectory, such as [7]. [8] also observed a quadratic relationship that follows a U-shaped curve between environmental deterioration and a nation's income level.

Despite widespread awareness, activism, and scientific evidence highlighting the dangers of carbon emissions, many countries and corporations face obstacles such as high costs, lack of technological expertise, financial constraints, competition from non-environmentally friendly companies, and inadequate infrastructure, preventing them from adopting a "zero emissions" policy [9–11].

This study has four distinct objectives 1. To understand the connection between $CO_2$ production, energy, income, and longevity across different regions. 2. To identify the drivers of carbon emissions and energy use in different regions and how they relate to income and life expectancy 3. To look into the synergies and trade-offs that might exist between various sustainability objectives, such as cutting carbon emissions, expanding access to energy, and enhancing health outcomes. 4. To assist in developing policies and strategies for decreasing carbon emissions and fostering sustainable development that are region specific. This study will add to the body of knowledge by offering a more nuanced understanding of the connection between $CO_2$ production, energy use, and longevity in different regions of the world. This can aid the identification patterns and drivers of carbon emission that are specific to different regions and inform the development of region-specific policies and strategies for reducing emissions.

## Review of relevant literature

This literature review addresses the complex dynamics of carbon production, income, energy consumption, and health across regions. It aims to fill the research gap by comparing different regions and identifying drivers of carbon production and energy usage, and their impacts on income and health. It also examines the effectiveness of policy interventions and technological advancements in reducing carbon emissions. The review focuses on recent peer-reviewed studies and is divided into three strands, each analyzing the impact of a variable on carbon emissions.

### Energy usage and carbon production

The academic community has extensively discussed the connection between energy use and carbon emissions. Studies show that energy consumption (gas, electricity, coal) generally has a

positive impact on $CO_2$, influenced by factors like environmental regulation and energy consumption structure.

To investigate their association [6] analyzed energy usage and economic development in the US from 1973 to 2015, various wavelet techniques were used. The findings revealed a favorable influence of energy usage on carbon production in the immediate, intermediate, and distant future. Causality tests confirmed that energy usage causes carbon production, emphasizing the need to reduce energy consumption to decrease carbon emissions. The US at the time aimed to attain a 50% decline in $CO_2$ productions and become a green economy by 2020 through promoting energy-efficient vehicles and implementing economic strategies to regulate polluting industries.

In another study [12] focused on drivers of $CO_2$ production in G7 countries over 27 years using panel cointegration methodologies. They found a constant and long-term association between carbon generation, trade, income, environmental innovation, and use of clean energy sources. Export, environmental innovation, and clean energy sources contribute to reducing consumption-driven $CO_2$ emissions, while imports and income increase consumption-driven $CO_2$ production. Policies targeting outward shipment, foreign purchases, income, and green technology have a substantial impact on carbon production. The study recommends eco-friendly approaches such as using green energy sources and promoting environmental innovation for achieving a sustainable environment.

Between 2005 and 2019 [13] studied the association between urbanization, energy utilization, economic expansion, and $CO_2$ production in African economies. Non-linear panel smooth transition regression demonstrated that energy use positively affects carbon production in both regimes, although an increase in urban population results in reduced emissions. Policymakers should enforce low-carbon policies to decrease energy usage and greenhouse gas emissions, while raising awareness of the hazards associated with carbon emissions.

Using data from 72 panels collected over time, [14] explored the connection between sustainable energy, non-sustainable energy, and carbon output. The outcomes showed integrated influences over time and highlighted the positive effect of non-renewable energy on lowering environmental dangers. Environmental deterioration is negatively and significantly impacted by economic expansion.

In a related vein [15] investigating sustainable energy's effect on $CO_2$ production, a panel cointegration study over a 23-year period in 107 nations found a negative association between sustainable energy and carbon emissions in high-income countries and a positive correlation in low-income countries.

Likewise, the study conducted by [16] delves into carbon emissions among countries with the potential to become major economies in the 21st century (N-11) from 1990 to 2017. This investigation meticulously examines unique variables such as financial development, human capital, renewable energy utilization, and GDP as potential influencers of $CO_2$ emissions. Employing diverse analytical methodologies, the study illuminates a positive connection between carbon emissions and both financial development and GDP. Conversely, it reveals an inverse correlation between carbon emissions and both technological innovation and the uptake of renewable energy sources. The study underscores the urgency of prioritizing technological innovation and the incorporation of renewable energy to align with the goals outlined in COP21.

The impact of agricultural practices, globalization, and renewable energy on ecological footprint and $CO_2$ emissions was investigated by [17]. The study focused on BRIC nations during the period from 1971 to 2016. The researchers discovered a lasting connection between Brazil and China. In China, renewable energy mitigated environmental strain, while in Brazil, it improved environmental quality. The study also revealed correlations between agricultural

activities and environmental degradation, as well as between globalization, ecological footprint, $CO_2$ emissions, and renewable energy generation. The findings emphasize the necessity of diverse strategies for renewable energy production to achieve sustainable development objectives.

Again, [18] investigating the effects of sustainable energy and healthcare spending on infrastructural capacity in Japan and the USA from 1982 to 2016, found clean energy and healthcare spending to improve environmental quality in the USA, while their impact on the load capacity factor in Japan was negligible. Economic growth significantly contributed to environmental degradation in both countries, highlighting the need for green growth, increased healthcare spending, and diversified sustainable energy sources to reduce environmental degradation.

## Economic growth and carbon production

A survey of research on economic progress and carbon emissions suggests that while economic progress typically increases carbon emissions, certain nations experience a decrease in emissions due to factors like technological innovation and changes in GDP composition. In China, a study using the augmented ARDL approach found that the Environmental Kuznets Curve hypothesis does not hold true. Instead, a U-shaped quadratic relationship exists between environmental degradation, income level, $CO_2$ emissions, and ecological footprint. Global integration, free trade, and income contribute to environmental pollution, while human capital reduces ecological footprint in the long run. The study emphasizes the importance of human capital in reducing environmental degradation in China, highlighting the insufficiency of clean energy utilization alone [8].

Focusing on Africa's growth diversity, [19] examined the impact of institutional changes, technological improvements, and sustainable energy sources. Using the Augmented Mean Group Estimator, it was found that both upper middle-income nations (UMICs) and lower middle-income countries (LMICs) experienced minimal effects of renewable energy sources on economic progress. Conversely, lower-income nations (LICs) experienced a detrimental effect.

Contributing to literature on $CO_2$ emissions and their impact [20] analyzed 28 African countries over 29 years, they investigated the impact of energy consumption on carbon production and the moderating role of per capita GDP. The findings indicated that increased energy consumption exacerbates carbon production, with per capita GDP acting as a positive driver of emissions. However, per capita income weakens the effects of energy utilization on emissions. Significant differences were observed across sub-regions, with Southern Africa contributing the most to emissions from energy consumption and Central Africa having the greatest detrimental effects through an increase in per capita GDP. West Africa showed the largest interaction effect. The study highlights the challenge for African economies in pursuing development and suggests prioritizing investments in sustainable energy to mitigate carbon emissions.

Also, [21] used fixed effects regression and threshold effects regression, to study 139 countries to understand the factors affecting carbon production, including emission efficiency, income inequalities, old age, and economic growth. The findings revealed that economic disparity has a negative impact on increasing carbon efficiency.

In a similar vein [22] examined the association between carbon production and income inequalities in the United States using a smooth changing coefficient model and a bandwidth regression model to achieve their aim. The findings suggest that higher inequality was initially linked to decreasing $CO_2$ production but increased towards the end of the sample period.

In contrast to consumption-based household carbon emissions, [23] utilized input-output analysis and bi-proportional scaling methods to achieve consumption-based and income-based annual accounting of urban and rural household emissions. Decomposition analysis was conducted to identify the main contributing factors. The findings showed that income-based household emissions constituted a larger proportion of total emissions compared to consumption-based household emissions. Income and expenditure were identified as the primary drivers of per capita household emissions.

Examining the non-linear relationship between tourism enhancement, economic progress, urban sprawl, and environmental deterioration in top tourist destinations, [24] used panel smooth transition regression (PSTR) with two regimes over 22 years. The results revealed a non-linear trajectory between tourism enhancement and environmental deterioration, with a negative impact above the threshold level and a positive impact below it. An inverted U-shape relationship was observed between tourism development and environmental deterioration, where initial increases in tourism enhancement led to environmental degradation but eventually decreased it. Economic development and urban sprawl also followed a non-linear trajectory dependent on the regime. The study provides valuable insights for policies and empirical research on sustainable tourism development. In summary, these studies collectively underscore the intricate relationships between economic progress, carbon emissions, and environmental factors. They highlight the multifaceted nature of these interactions, offering insights into effective policy strategies and areas for further research in achieving sustainable development.

## Mortality and carbon production

Numerous studies have examined the impact of carbon emissions on health outcomes, highlighting adverse effects on cardiovascular issues, lung cancer, respiratory diseases, and increased mortality, particularly in highly industrialized and urbanized areas.

To that point, [25] analyzed the relationship between carbon production, life expectancy, and mortality by considering different sources of carbon production and degrees of economic development. The findings showed a negative association between $CO_2$ emissions and mortality.

Furthermore, a separate study conducted by [26] delved into the reciprocal relationship between air pollution and health outcomes. This study was unique in its methodology, enabling a clear distinction between observed health effects and potential health impacts. Longitudinal data from 29 European countries were employed to explore this dynamic. The findings from this research reveal, among other insights, that pollutants impacting European nations possessed the potential to adversely affect children at birth.

Using life expectancy as a proxy for mortality [27] investigated the connection that exist in Pakistanis life expectancy and their economic growth. The Autoregressive Distributive Lag (ARDL) Bounds Test was employed to meet their goals. The outcomes supported the existence of long-term relationship among the variables. The findings included, among other things, that life expectancy is decreased by energy consumption as a result of environmental degradation.

To mitigate carbon emissions, it is crucial to comprehend the factors driving them. In this regard, a study by [28] investigated the determinants of carbon emissions in the ten highest $CO_2$-producing countries. The researchers employed the Logarithmic Mean Divisia Index (LMDI) to dissect the contributors to carbon emission changes, including population, energy intensity, carbon intensity impacts, and per capita income. The study revealed that, for the majority of the countries under scrutiny, variables such as the Human Development Index

(HDI), economic growth, and mortality were intricately and significantly intertwined with carbon emissions.

Examining health consequences of carbon production in the context of global warming in China. [29] used regression analysis on provincial-level panel data for 15 years. The research discovered that carbon production has a negative and long-lasting impact on the health of inhabitants, primarily by increasing temperatures. In regions with high levels of industrial development and extensive urban growth, heightened carbon production presents immense health hazards. The study proposes that enhancing industry nature, enhancing proper urban structures, and promoting synchronized industrial growth and urban progress can decrease the harmful effects of carbon production on inhabitants' health in China's distinct "leading industrialization and lagging urbanization" condition. The study highlights that a standardized policy model is unsuitable for China's present situation, and mitigation measures must be tailored to local conditions in economically developed or less developed areas. The authorities must concentrate on the interaction and synergy between industrialization and urbanization to address global climate change issues while considering the health hazards posed by carbon emissions.

Similarly, [30] revealed that temperature in the environment has a substantial impact on mortality, particularly in susceptible populations like the young, old, and those with co-morbid diseases. The number of fatalities rises as temperatures go above a specific threshold. This problem is made worse by global warming, which will also negatively affect primary producers and livestock, resulting in undernourishment and health problems. To lower the carbon footprint and prevent global warming, public health initiatives should also include enhanced urban design, energy consumption reduction, and health-related information efforts.

Utilizing insights from the reviewed literature strands, it becomes apparent that energy usage significantly contributes to carbon emissions, consequently yielding adverse health effects. However, the relationship between carbon emissions and income is intricate and subject to regional variations. To the best of our knowledge, a comprehensive study examining the effects of carbon emissions across regions with diverse income levels and conditions is conspicuously absent in the existing body of knowledge. To address this research gap, the present study undertakes an investigation into the intricate interplay among carbon emissions, energy consumption, income, and health outcomes across heterogeneous regions. Through this endeavor, the research aspires to provide invaluable insights for both policymakers and researchers.

## Materials and methods

### Description of variables

The study uses a range of variables to explore the association between environmental deterioration, healthier life, longevity, and economic development. Carbon emission per capita is used to represent environmental deterioration, while life expectancy and death rate serve as indicators of healthier life and longevity. Energy consumption is considered as a contributing factor to carbon emission, and per capita GDP is used to represent economic growth. The study also includes total population as the control variable in the analysis. The data used in the study are sourced from the World Bank Data, and Table 1 provides a clear description of the variables used and their respective units of measurement. Overall, the selection of variables and their operationalization is based on existing literature and provides a robust framework for examining the relationship between environmental degradation, healthier life, longevity, and economic growth.

**Table 1. Description of variables.**

| VARIABLES | CODE | UNIT OF MEASUREMENT |
|---|---|---|
| Carbon emissions | Carbon | Metric tons per capita mt/pc |
| Life expectancy | Life | Years |
| Energy consumption | Energy | Oil equivalent in Kg |
| Crude death rate | Death | Per 1000 people |
| Per capita GDP | GDPpcap | Constant US$ |
| Population | Pop | Thousands, millions, billions |

## Theoretical rationale and specifying empirical model

To achieve our study objectives, we utilized four models based on previous research [28, 31]. The first model examined the relationship between carbon emissions and life expectancy, assessing the impact of life expectancy changes on carbon production. The second model explored the link between carbon emissions and death rates, investigating how fluctuations in death rate affect energy production. The third model investigated the effects of carbon emissions on life expectancy, shedding light on environmental factors' implications for longevity. The fourth model examined the presence of a Kuznets curve, exploring the relationship between carbon emissions, economic development, and sustainability. Additionally, a fifth model investigated the presence of an inverted N-curve, indicating a non-linear relationship between emissions and economic development.

Previous studies [4, 7] have explored these complex relationships. The models represented variables such as Carbon (carbon emissions per capita), Energy (energy consumption), Pop (total population), and GDPpcap (per capita GDP) using natural logarithms. The regression models were specified accordingly.

$$lnCARBON_{i,t} = \alpha + \beta_{11}lnLIFE_{it} + \beta_{12}lnENERGY_{it} + \beta_{13}lnPOP_{it} + \beta_{14}lnGDPpcap_{it} + \phi_{it} + \varepsilon_{it} \quad (1)$$

$$lnCARBON_{i,t} = \alpha + \beta_{21}lnDEATH_{it} + \beta_{22}lnENERGY_{it} + \beta_{23}lnPOP_{it} + \beta_{24}lnGDPpcap_{it} + \phi_{it} + \varepsilon_{it.} \quad (2)$$

$$lnLIFE_{it} = \alpha + \beta_{31}lnCARBON_{it} + \beta_{32}lnENERGY_{it} + \beta_{33}lnPOP_{it} + \beta_{34}lnGDPpcap_{it} + \phi_{it} + \varepsilon_{it.} \quad (3)$$

$$lnCARBON_{i,t} = \alpha + \beta_{41}lnLIFE_{i,t} + \beta_{42}lnENERGY_{i,t} + \beta_{43}POP_{i,t} + \beta_{42}lnGDPpcap_{it}$$
$$+ \beta_{42}lnGDPpcap^2_{i,t} + \phi_{i,t} + \varepsilon_{i,t}. \quad (4)$$

$$lnCARBON_{i,t} = \alpha + \beta_{51}lnLIFE_{i,t} + \beta_{52}lnENERGY_{i,t} + \beta_{53}POP_{i,t} + \beta_{54}lnGDPpcap_{it}$$
$$+ \beta_{55}lnGDPpcap^2_{i,t} + \beta_{56}lnGDPpcap^3_{i,t} + \phi_{i,t} + \varepsilon_{i,t}. \quad (5)$$

## Estimation technique

The investigation utilizes panel data, offering advantages such as examining groups instead of individual panels and minimizing information loss. To ensure statistical inferences, various tests were conducted. First, the Fisher-type and Im-Pesaran-Shin (IPS) tests checked variable stationarity for unbalanced panel data with cross-sectional dependence (CSD). Second, tests for cross-sectional dependence, cointegration (Kao and Westerlund), and AR disturbances

(Wooldridge) were performed. The IPS unit root test, suitable for our linear trend assumption, accounts for unique unit characteristics. The equation provided by [32] captures these considerations.

$$\sqrt{N \frac{\overline{t}N, t - \mu}{\delta}} \Rightarrow N(0, 1),\ \overline{t}_{N,T} = \frac{1}{N} \sum_{i=1}^{N} t_{i,T}. \tag{6}$$

The equation represents the asymptotic distribution of the IPS test statistic in the unit root hypothesis of panel data. It calculates the average of individual unit root test statistics at time t, denoted by $\overline{t}$-N,T, along with the mean ($\mu$) and variance ($\delta$) of these averages. Estimating $\mu$ and $\delta$ in the IPS test involves Monte Carlo methods. However, the IPS test assumes constant T across all cross-sectional units, implying uniform E (ti, T) and V (ti, T) for all units. Unbalanced panel data, like in our research, may require additional simulations for reliable results. Nevertheless, the IPS test offers a robust approach for analyzing panel data and understanding unit root dynamics.

$$CD = \sqrt{\frac{2T}{N(N-1)}} \left( \sum_{i=1}^{N-1} \sum_{J=1+1}^{N} \hat{p}_{i,j} \right) \tag{7}$$

Eq 7 tests cross-sectional dependence in panel data analysis using the pair-wise correlation coefficient between units i and j. The Kao and Westerlund cointegration tests, applied to numerous individual units with many observations, offer a powerful approach by combining statistics for each member of the panel [33, 34]. The combined test has a standard normal distribution after sufficient standardization, unlike single time series cointegration tests with non-standard distributions.

$$y_{i,t} = x'_{i,t}\beta_i + \acute{Z}_{i,t}\gamma_i + e_{i,t} \tag{8}$$

For the I (1) dependent variable $y_{i,t}$, all tests in xtcointtest are based on a panel-data model, where i = 1,. . .,N signifies the panel (individual) and t = 1,. . .,T denotes time. The covariates must not be cointegrated among themselves for the tests to be valid. Panel-specific means, and panel-specific temporal trends, or nothing are all possible using the vector $Z_{i,t}$.

The null hypothesis across the tests is that $y_{i,t}$ and $x_{i,t}$ are not cointegrated. xtcointtest tests for no cointegration by testing that $\epsilon_{i,t}$ is nonstationary. Rejecting the null hypothesis requires that series $y_{i,t}$ and $x_{i,t}$ are cointegrated and that $\epsilon_{i,t}$ is stationary. The variables are cointegrated in all panels, which is the alternative hypothesis supported by the Kao tests, Pedroni tests, and all-panels Westerlund test. The cointegration of the variables in some of the panels is the alternative hypothesis for the some-panels variant of the Westerlund test. All tests permit imbalanced panels and condition that N be large enough for the sample average of panel-level statistics distribution to converge to the population distribution. Additionally, they demand that $T_i$ be big enough to support time series regressions with just that panel's observations.

According to Wooldridge's method, the individual-level effect is eliminated from a regression model using first differences, and the parameters are then estimated by regressing $\Delta y_{i,t}$ on $\Delta X_{i,t}$ and obtaining the residuals $\epsilon_{i,t}$. Corr ($\Delta_{it}$, $\Delta_{i,t-1}$) = -0.5 if the residuals are not serially correlated. The process then performs a regress on the residuals' lagged values and checks to see if the coefficient is equal to -0.5. The VCE is modified for clustering at the panel level to take into

account within-panel correlation. This test is also robust to conditional heteroskedasticity.

$$y_{it} - y_{it-1} = (X_{i,t} - X_{it-1})\beta_1 + \epsilon_{i,t} - \epsilon_{i,t-1} \tag{9}$$

$$\Delta y_{i,t} = \Delta X_{i,t}\beta_1 + \Delta\epsilon_{i,t} \tag{10}$$

where $\Delta$ is the first-difference operator.

To address the issues of panel data, we employed the Feasible Generalized Least Squares (FGLS) and Panel Corrected Standard Errors (PCSE) techniques, which assume heteroscedastic and contemporaneously correlated disturbances across panels. We assumed first-order autocorrelation and panel-level heteroscedastic errors and included all available observations with non-missing pairs. The Prais-Winsten PCSE was used to limit statistical over-confidence, and FGLS was used as the primary estimation technique. The robustness of these techniques has been demonstrated in previous research works [26, 35]. The use of PCSE ensures that standard errors are resilient to geographical and temporal dependency, even when the time dimension is large. The number of panels does not pose a barrier to viability since the size of the cross-sectional dimension does not affect the limiting behavior of the number of panels in this non-parametric method of computing standard error.

## Pre-estimation diagnostics

To ensure the reliability of our findings, we examined cross-sectional dependence among the variables [36]. Except for per capita GDP constant (GDPpcap), which exhibited cross-sectional dependence, all other variables were cross-sectionally independent. This dependence in GDPpcap may be influenced by external factors like global economic conditions, trade policies, and political stability [37–39]. We employed statistical techniques to address this issue. Cointegration tests were conducted to identify long-term relationships and common stochastic trends among the non-stationary variables. Both the Kao and Westerlund cointegration tests confirmed the presence of long-run relationships among the variables. Table 2 presents the cross-sectional dependence test (CSD), while Table 3 displays the results of the cointegration tests.

Our pre-estimation tests also included unit root tests, we used both first generation and second-generation tests, that is the augmented Dickey-Fuller unit root test (ADF) and the Im-Pesaran and Shin tests (IPS).

The IPS which is considered a second-generation unit root test was developed in response to some of the limitations of first-generation unit root tests. The test uses a more efficient estimator of the autoregressive parameter and allows cross-sectional dependence in the data. It

**Table 2. Pesaran cross-sectional dependence test.**

| VARIABLE | CD-TEST | P-VALUE | AVERAGE JOINT T | MEAN ρ | MEAN abs (ρ) |
|---|---|---|---|---|---|
| CARBON | .067 | 0.947 | 27.03 | 0.00 | 0.16 |
| LIFE EXPECTANCY | -.802 | 0.423 | 30.83 | 0.00 | 0.15 |
| CRUDE DEATH RATE | -.476 | 0.634 | 30.92 | 0.00 | 0.15 |
| ENERGY USE | -.534 | 0.594 | 18.95 | 0.00 | 0.14 |
| POPULATION | .027 | 0.787 | 30.75 | 0.00 | 0.15 |
| GDP PER CAPITA | 195.654 | 0.000 | 29.32 | 0.53 | 0.71 |

Notes: Under the null hypothesis of cross-section independence, CD ~ N(0,1)

P-values close to zero indicate data are correlated across panel groups.

**Table 3. Kao and Westerlund cointegration test.**

|  | Test statistic | P-value |
|---|---|---|
| Modified Dickey-Fuller t | -11.4881*** | 0.0000 |
| Dickey-Fuller t | -20.0047*** | 0.0000 |
| Augmented Dickey-Fuller t | -8.1726*** | 0.0000 |
| Unadjusted modified Dickey-Fuller t | -33.3915*** | 0.0000 |
| Unadjusted Dickey-Fuller t | -27.4460*** | 0.0000 |
| **Westerlund cointegration** | | |
| Variance ratio | -4.7645*** | 0.0000 |

Ho: No cointegration

Ha: All panels are cointegrated

has been shown to have good power and properties in simulations and widely in empirical research. To keep things concise, we will only present the IPS table in this research, while the ADF table will be included in the supplementary data. Both tests revealed that at their levels all variables were stationary across a deterministic mean except GDPpcap which became stationary after log transformation and including a lag with cross-sectional means.

Autocorrelation test was conducted using the Wooldridge test for AR1 disturbances, we opted for this test because of its good power and size properties, meaning it can detect first-order autocorrelation with high accuracy while controlling the probability of a false positive (type 1 error). The test output indicated that our model suffers from first-order autocorrelation. Table 4 displays the results of the unit root and Woodridge autocorrelation tests. We employed the PCSE and FGLS to deal with the deficiencies of panel data. To deal with autocorrelation the PCSE method uses the variance-covariance matrix estimator that accounts for the presence of autocorrelation in the errors. This estimation is based on the modified version of the White estimator, which is a consistent estimator of the variance-covariance matrix in the presence of heteroscedasticity but not autocorrelation. The modified estimator used in PCSE method adjusts for both heteroscedasticity and autocorrelation by estimating the variance-covariance matrix separately for each individual unit and then taking the average across units. This accounts for both within-unit and between-unit correlations in the errors. The PCSE can

**Table 4. Im-Pesaran and Shin unit root tests and autocorrelation test.**

| variables | Levels statistic | p-value | Log transformation Statistic. Demean, trend lag (1) | p-value | First differences | p-value |
|---|---|---|---|---|---|---|
| GDPpcap | 9.4936 | 1.0000 | -7.5057 | 0.0000 | - | - |
| Carbon | -26.6781 | 0.0000 | -17.3524 | 0.0000 | - | - |
| Life expectancy | -29.7440 | 0.0000 | -22.1210 | 0.0000 | - | - |
| deathrate | -30.1564 | 0.0000 | -23.9121 | 0.0000 | - | - |
| Energy use | -17.2856 | 0.0000 | - | - | - | - |
| population | -27.0998 | 0.0000 | -20.3074 | 0.0000 | - | - |
| **Wooldridge test for autocorrelation in panel data** | | | | | | |
| Test | Test statistic | P-value | Decision criteria | | | |
| Wooldridge test for AR 1 disturbances | 7.810 F (1, 80) | 0.0065 | Reject the null hypothesis | | | |

Null hypothesis for Wooldridge test

$H_0$: No autocorrelation

improve the efficiency and accuracy of regression coefficient estimates in panel data models. The FGLS uses a weighted least squares approach, where the observations are weighted according to their estimated variance. The weights are based on their consistent estimator of the variance-covariance matrix of the errors, such as white estimator or Newey-West estimator. To deal with autocorrelation the FGLS estimator uses generalized least squares approach, where the errors are assumed to follow a specific structure. The covariance structure is typically based on first-order autoregressive process, which assumes that the current error is correlated with previous error with a specific lag. The FGLS estimator estimates both regression coefficients and the parameters of the covariance structure simultaneously using maximum likelihood structure [40].

## Description of data

This study utilized unbalanced panel data for 82 countries, categorized into three regions (ECA, MENA, SSA) based on the World Bank's classification. The dataset covered the period from 1990 to 2020. Summary statistics (presented in Tables 5 and 6) were calculated for the combined sample and each region separately.

For carbon emissions, there were 2,779 observations, ranging from 0 to 47.669 mt/pc, with a mean of 5.413 and a dispersion of 6.2449. Life expectancy had a mean of 68.01636, a standard deviation of 10.82, and ranged from 26.173 to 83.9. The crude death rate had a mean of 9.8, a standard deviation of 4.25, and ranged from 1.127 to 41.359 deaths. Energy consumption included 1,988 observations, with a mean of 159.1407, a standard deviation of 120.584, and ranged from 39.43401 to 990.0784. The variable GDP per capita constant was analyzed using a sample of 2,891 observations.

The results showed that the mean GDP value was 13,814.51, with a relatively high standard deviation of 19,333.76. The minimum and maximum GDP values of 190.2342 and 112,417.9, respectively.

The present study analyzed the variable total population using a sample of 2,964 observations. The mean value of the population was found to be 17,500,000, with a standard deviation of 25,300,000. These statistics indicated significant variations across panels, highlighting the wide range of values for each variable and the high degree of variability and dispersion within the data

Our analysis has identified the countries that emit the most carbon dioxide in descending order. Qatar has the highest average annual emission of 35.6171 mt/pc, followed by the United Arab Emirates with an emission of 25.034123 mt/pc. Kuwait is the third highest emitter with 23.66mt/pc average emissions per year, followed by Bahrain with 22.66 mt/pc, and lastly Luxembourg with 21.81 mt/pc average emissions. Notably, all of the top four highest emitters are

**Table 5. Summary statistics of MENA, SSA, and ECA regions combined (Total Sample).**

| Variable | observations | Mean | Standard deviation | Minimum | maximum |
|---|---|---|---|---|---|
| Carbon emissions | 2,779 | 5.413257 | 6.244919 | 0 | 47.69993 |
| Life expectancy | 2,968 | 68.01636 | 10.8243 | 26.172 | 83.90488 |
| Death rate | 2,972 | 9.821474 | 4.256023 | 1.127 | 41.359 |
| Energy Use | 1,988 | 159.1407 | 120.584 | 39.43401 | 990.0784 |
| GDP per capita | 2,891 | 13814.51 | 19333.76 | 190.2342 | 112417.9 |
| Population | 2,964 | 1.75e+07 | 2.53e+07 | 254826 | 2.06e+08 |

Source: Authors' computations

Note: ECA = Europe and Central Asia, MENA = Middle East and North Africa, SSA Sub-Saharan Africa.

**Table 6. Region-wise summary statistics.**

| Variable Region: ECA | Observation | Mean | Standard deviation | Minimum | Maximum |
|---|---|---|---|---|---|
| Carbon | 1,363 | 6.880647 | 3.909361 | .3214923 | 30.43928 |
| Life expectancy | 1,449 | 74.78009 | 5.125184 | 58.104 | 83.90488 |
| Death rate | 1,453 | 9.949971 | 2.568571 | 4.6 | 18 |
| Energy use | 1,152 | 162.4228 | 127.3303 | 39.43401 | 815.9927 |
| GDP per capita | 1,152 | 21447.46 | 22331.94 | 371.8312 | 112417.9 |
| Population | 1,457 | 1.86e+07 | 2.77e+07 | 254826 | 1.49e+08 |
| **Region: MENA** | | | | | |
| Carbon | 460 | 10.76723 | 10.29498 | .8759462 | 47.69993 |
| Life expectancy | 496 | 74.19336 | 3.701276 | 64.732 | 82.85854 |
| Death rate | 496 | 4.330645 | 1.697889 | 1.127 | 8.1 |
| Energy use | 349 | 127.9936 | 59.32339 | 47.85092 | 412.0265 |
| GDP per capita | 470 | 17066.68 | 17160.06 | 726.7388 | 73493.27 |
| Population | 493 | 1.19e+07 | 1.20e+07 | 354170 | 4.39e+07 |
| **REGION: SSA** | | | | | |
| Carbon | 956 | .744974 | 1.41831 | 0 | 8.568994 |
| Life expectancy | 1,023 | 55.44117 | 7.390869 | 26.172 | 74.51463 |
| Death rate | 1,023 | 12.30119 | 4.590986 | 5.047 | 41.359 |
| Energy use | 487 | 173.698 | 132.6519 | 57.76555 | 990.0784 |
| GDP per capita | 1,012 | 1676.804 | 1942.557 | 190.2342 | 10959.34 |
| Population | 1,014 | 1.87e+07 | 2.60e+07 | 949493 | 2.06e+08 |

Source: Authors' computations

Note: ECA = Europe and Central Asia, MENA = Middle East and North Africa, SSA Sub-Saharan Africa.

located in the Middle East, while the fifth highest emitter comes from Europe. On the other hand, Burundi (0.03560219 mt/pc), Central African Republic (0.06136718 mt/pc), Rwanda (0.06907359 mt/pc), and Chad (0.06933901 mt/pc) are the least emitters of carbon in our sample, arranged in descending order. Interestingly, all the countries with the least carbon emissions occur in sub-Saharan Africa. These findings provide insights into the distribution of carbon emissions across different regions and can inform policies and strategies aimed at mitigating climate change and reducing carbon footprints.

The country with the highest energy consumption in the sample is Turkmenistan, with an average of 632.47 kg, followed by Mozambique in sub-Saharan Africa with 617.40 kg, and Uzbekistan with 565.50 kg. Azerbaijan and Ukraine are the countries with the lowest energy consumption levels, with figures of 299.35457 kg and 565.50133 kg, respectively. With the exception of Mozambique, all the countries with high energy consumption are located in Europe and Central Asia. On the other hand, Switzerland, Mauritius, Italy, Turkey, and Ireland are the countries with the lowest energy consumption levels, with respective figures of 57.59 kg, 68.66 kg, 69.04 kg, 72.94 kg, and 73.11 kg. The least energy consumers are also found in Europe and Central Asia and sub-Saharan Africa. These findings suggest that energy consumption levels vary widely across different regions and may be influenced by various factors such as economic development, energy policies, and environmental awareness.

The regions with the highest population sizes are Russia Federation (145,300,000), Nigeria (143,300,000), Germany (81,794,325), Turkey (68,304,756), and France (63,006,742). In contrast, the least populated countries are Iceland (301,418.52), Malta (411,133.39), Luxembourg (482,551.45), Bahrain (973,069.19), and Cyprus (1,014,087.7). Notably, except for Nigeria,

which is located in sub-Saharan Africa, the regions with the highest and least populated countries are in Europe and Central Asia. These findings suggest that population distribution across different regions is not evenly distributed and may be influenced by various factors such as economic development, migration patterns, and political stability.

Sub-Saharan Africa has the lowest life expectancy of all regions in our sample, with the top five countries with the least life expectancy located in this region. Sierra Leone has the lowest life expectancy at 45 years, followed by the Central African Republic at 47 years, Nigeria at 49.1 years, Chad at 49.76 years, and Zimbabwe at 51 years. In contrast, the countries with the highest life expectancy are located in Europe, with Switzerland having the highest average life expectancy of 80.93 years, followed by Iceland at 80.85 years, Italy at 80.60 years, Sweden at 80.46 years, and Spain at 80.40 years. These findings highlight the significant disparities in life expectancy across different regions and can inform policies and interventions aimed at improving healthcare and reducing health inequalities.

Burundi has the lowest GDP per capita in our sample, at 193.60 USD, followed by Malawi at 319.98 USD, Sierra Leone at 345.94 USD, Eritrea at 364.54 USD, and Niger at 386.82 USD. Significantly, all these countries are located in sub-Saharan Africa. In contrast, Luxembourg records the highest GDP per capita of 80,522.98 USD, followed by Switzerland, Norway, Denmark, and Qatar in descending order, with figures ranging from 62,032.183 USD to 44,054.41 USD. All these countries are located in the Europe and Central Asia region. These findings highlight significant disparities in economic productivity across different regions and can inform policies and strategies aimed at reducing economic inequalities and promoting sustainable economic growth.

Our final variable is the death rate, and the countries with the lowest death rates are Qatar with 1.59, United Arab Emirates with 1.71, Kuwait with 2.51, Bahrain with 2.66, and Oman with 2.35. On the other hand, countries with the highest death rates are located in sub-Saharan Africa, with Sierra Leone having the highest crude death rate, followed by the Central African Republic, Rwanda, Chad, and Nigeria in descending order, with corresponding figures of 19.18, 16.6, 15.97, 15.89, and 15.74.

We plotted the relationships between the variables in our models. To this end, we conducted a preliminary analysis of the associations between various indicators of economic growth, carbon emissions, life expectancy, and death rates across countries in different regions. Our findings revealed in Fig 1 a quadratic relationship between GDP per capita and carbon emissions, with an upward trending linear fit line. Specifically, we observed that carbon production exhibited an increasing relationship with GDP per capita, while the quadratic fit line showed an inverted U-relationship, suggesting the existence of the Kuznets curve. Further analysis is needed to explore this phenomenon in greater depth.

Moreover, we found a linear association between life expectancy and per capita GDP, with an asymptotic increase in life expectancy as income rises. Interestingly, countries with higher life expectancy tended to have higher incomes. This relationship is indicated in Fig 2. Furthermore, our analysis indicated a positive relationship between life expectancy and carbon emissions, with an upward trending linear fit line. Notably, the region of most interest in this graph was the far right-hand corner, above 75 years of life expectancy but under 1 mt/pc of carbon emissions per capita. This area suggests environmental sustainability in terms of high life expectancy. Further exploration revealed that countries in the ECA occupy this region. The association is illustrated in Fig 3. Finally, we observed a negative relationship between carbon emissions and death rates, with a downward trending linear fit line. Notably, most countries were clustered around low death rates and low carbon emissions, warranting further investigation. The results are presented in Fig 4. The present study employed choropleth maps to visualize the mean distribution of life expectancy and carbon emission per capita across various

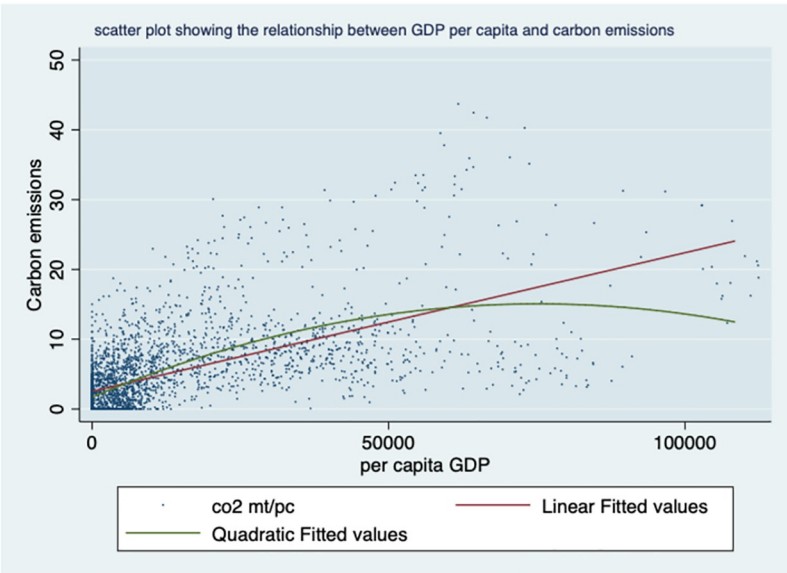

**Fig 1. Scatter plot showing the relationship between GDP per Capita and carbon production.** Source: World Bank data indicators.

countries. Figs 5 and 6 depict the geographical distribution of life expectancy and carbon emission per capita, respectively.

Our analysis revealed that countries shaded red in the life expectancy map had the lowest life expectancy, with ages ranging from 39 to 53. Significantly, most of these countries were found in the SSA region, while Russia, which is located in the ECA region, recorded low figures of life expectancy. Conversely, areas color coded deep blue on the life expectancy map represented countries with high life expectancy within the range of 72.48–80.79. Most of these countries were found in the ECA region. Moreover, Fig 6 showed that areas color coded red

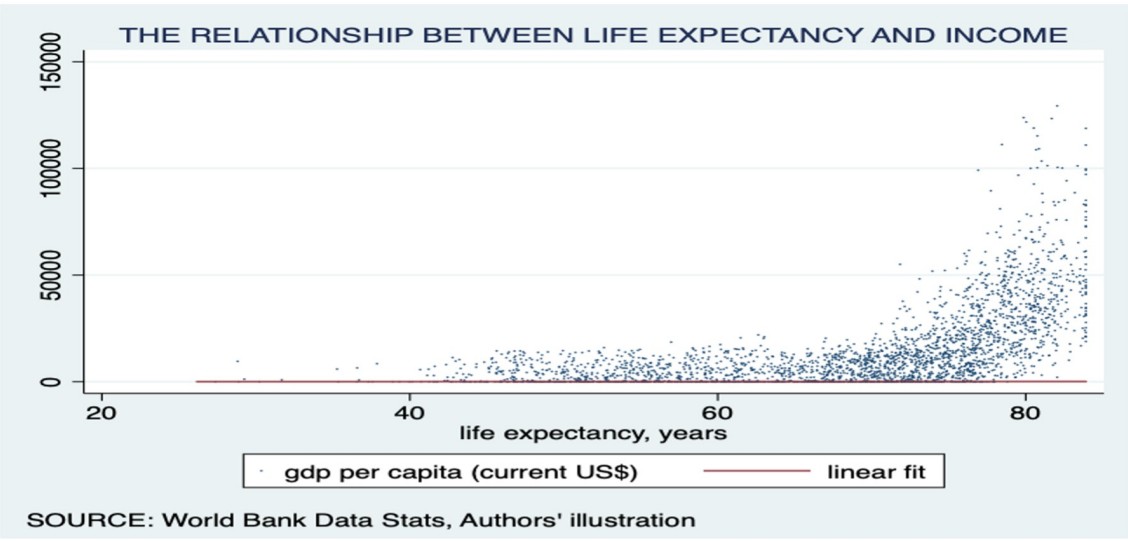

**Fig 2. Scatter plot showing the association between life expectancy and GDP per Capita.** Source: World Bank data indicators.

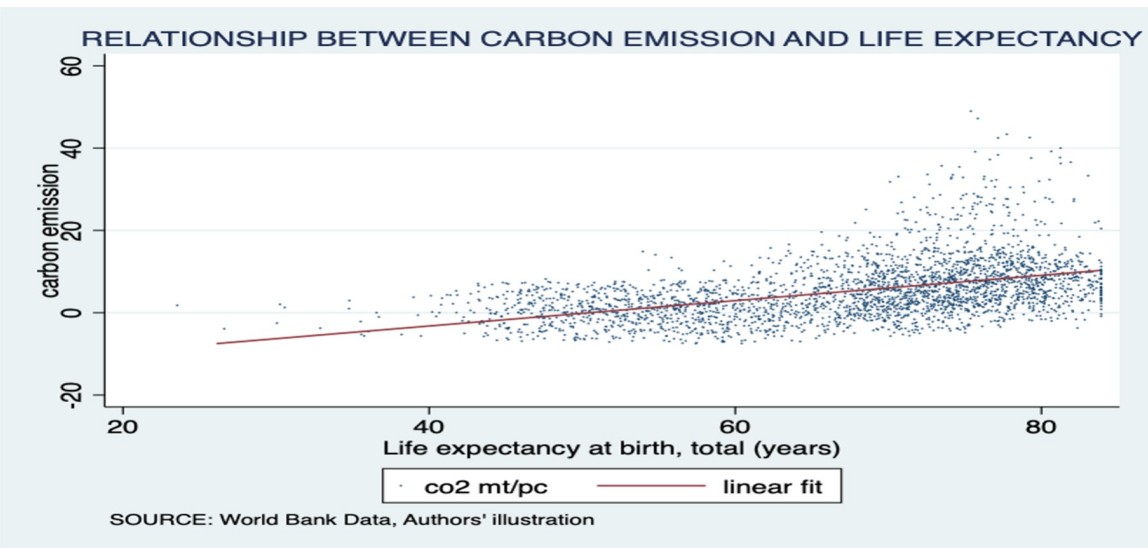

**Fig 3. Scatter plot depicting the association between carbon production and life expectancy.** Source: World Bank data indicators.

on the carbon emission map illustrated countries with carbon pollution above 6.947 mt/pc. Conversely, countries in the lighter blue regions emitted the least amount of carbon emissions, ranging from 0.034488 to 0.342563. These findings highlight the significant disparities in environmental sustainability across different regions of the world.

Overall, our findings highlight the complex interplay between economic growth, environmental sustainability, and human well-being. Further research is needed to explore these relationships in greater depth and to develop policies that promote sustainable development and

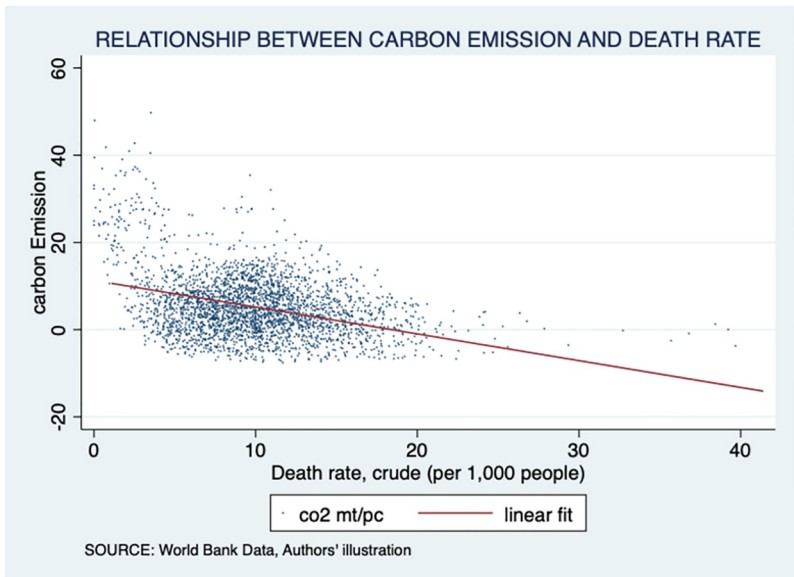

**Fig 4. Scatter plot showing the association between carbon production and death rate.** Source: World Bank data indicators.

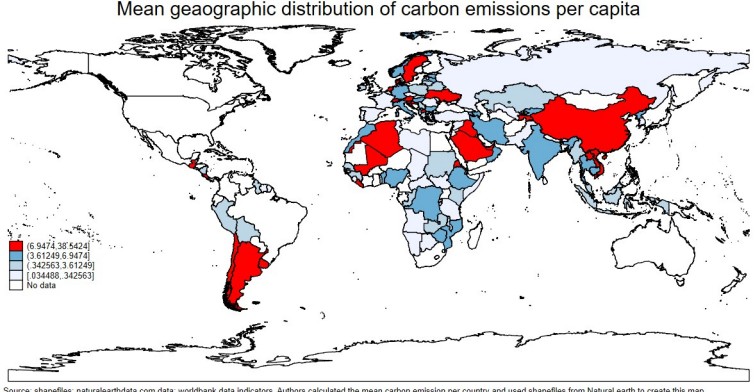

**Fig 5. Choropleth map illustrating the mean geographic distribution of life expectancy.** Source: shapefiles: Naturalearthdata.com, graph: Authors' illustration.

human flourishing. These findings provide insights into the significant variations in health outcomes across different regions.

## Results and discussion

### Results for Middle east and North Africa, Europe and Central Asia, and sub-Saharan Africa combined (Total Panel Estimation)

The study used FGLS and PCSE panel fixed effects to estimate the results of three models, and the findings are presented in Table 7. The authors followed a technique developed by [41], which involved considering fixed effects specific to each country and time effects across countries. All variables in the three models were statistically significant at a 1% level, with predictive power ranging from 0.67 to 0.99, except for population. Population was found to be statistically insignificant in models 1 and 2, but it was significant in model 3. Time effects showed heterogenous parameters from 1990–2020. Confirming the presence of fixed effects and presence of heterogeneity. Furthermore, the intercept for models 1 and 2 is negative and significant whiles model 3 is positive and significant. Meaning that, for model 1 holding all other regressors constant carbon emissions increased with life expectancy by 2.92% -3.19%, energy by 0.471–

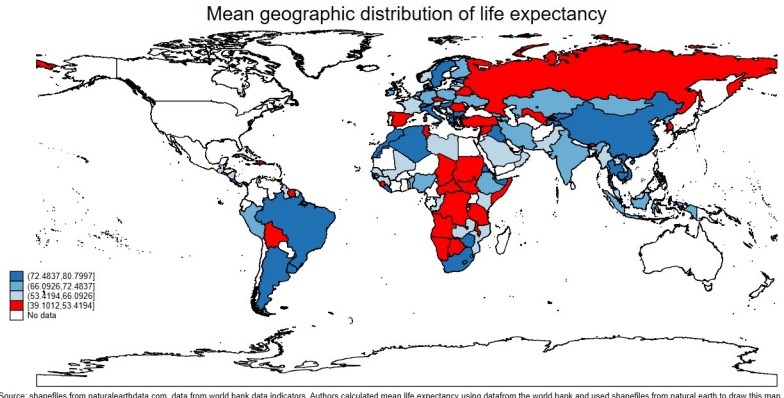

**Fig 6. Choropleth map illustrating the mean geographic distribution of carbon emission.** Source: shapefiles: Naturalearthdata.com, graph: Authors' illustration.

**Table 7. Results for Middle east and North Africa, Europe and Central Asia, and sub-Saharan Africa combined (Total Panel Estimation).**

| Variables | MODEL 1 | MODEL 1 | MODEL 2 | MODEL 2 | MODEL 3 | MODEL 3 |
|---|---|---|---|---|---|---|
| | PCSE | FGLS | PCSE | FGLS | PCSE | FGLS |
| lnCARBON | | | | | .059*** (20.56) | .052*** (28.06) |
| lnLIFE | 3.19*** (15.53) | 2.92*** (18.97) | | | | |
| lnDEATH | | | -.5434*** (-14.36) | -.546*** (-20.87) | | |
| lnENERGY | .57*** (17.28) | .471*** (17.58) | .440*** (14.12) | .426*** (22.40) | -.073*** (-18.46) | -.08*** -40.80 |
| lnPOP | -.012 (-0.96) | -.002 (-0.11) | .013 (0.93) | .001 (0.20) | .001*** 0.82 | .005*** 4.42 |
| lnGDPpcap | .66*** (31.38) | .64*** (28.05) | .845*** (44.08) | .725*** (56.38) | .02*** (6.59) | .015*** (7.92) |
| constant | -20.698*** (-23.36) | -19.05*** (-27.55) | -7.40*** (-18.78) | -5.94*** (-24.87) | 4.311*** (79.86) | 4.38*** (139.80) |
| No of Obs/grp | 1,933/82 | 1,933/ 82 | 1,937/82 | 1,937/82 | 1,933/82 | 1,933/82 |
| $R^2$/Common AR AR coefficient | 0.6963 | 0.7464 | 0.67 | 0.7231 | 0.99 | 0.6810 |
| Wald Statistic | 13618.50*** | 2317.19*** | 2565.28*** | 3666.40*** | 2512.95*** | 4167.10*** |
| Time dummies | Yes | Yes | Yes | Yes | Yes | Yes |

Source: authors' estimation

*** = 1% significance

** = 5% significance

* = 10% significance

() = Z-statistics

0.57%, GDPpcap by 0.64–0.66%%. For model 2, holding all other variables constant, a unit increase in death rate decreases carbon emission by -0.543% and -0.546, energy increases with carbon emission by 0.42 to 0.44% %, GDPpcap increases carbon emission by 0.725%-0.845%. In model 3 all the tested variables were found to be significant at the 1% level, with all variables held constant a unit increase in energy consumption will decrease carbon emissions by -0.073% -0.08%, population increases affected the increase in carbon emission by 0.001%-0.005%, and GDPpcap grew with carbon emission production by 0.015%-0.02%.

## Results for region-wise estimation

Upon conducting a regional analysis of the data, we observed that all the variables tested in the three different models were statistically significant at the 1% level, except for population in model 3 for ECA and SSA, and GDPpcap for MENA, which did not attain statistical significance. The coefficients and constants exhibited varying signs, but they were all statistically significant at the 1% level. Regarding model 1, when all other factors were held constant, carbon emissions rose by 1.35 percent for ECA, 1.40 percent for MENA, and 1.43 percent for SSA with rising life expectancy. A unit increase in the mortality rate will result in a 37% rise in carbon emissions, according to model 2's positive interaction between carbon emissions and ECA of 0.37%. On the other hand, for MENA and SSA, a unit rise in death rate will result in a 51% and 60% reduction in carbon emissions, respectively. Positive results for models 1 and 2's energy consumption suggested that energy use played a substantial role in the creation of carbon emissions in each of the three regions. However, Model 3 yielded negative results, indicating that increases in energy consumption reduced life expectancy in the regions under study.

In particular, the study discovered that increases in energy consumption increased carbon emissions in MENA by 53%-72% whereas they raised carbon emissions in ECA by 50%-58%. Energy use in SSA increased carbon emissions by 20%–20.9%, assuming constant conditions. Additionally, in Model 3 for ECA, increases in energy usage decreased life expectancy by 0.62%. While increases in energy consumption impacted life expectancy in SSA by 10%, they

did so in MENA by a much smaller 0.5%. Further investigation revealed that population growth had a significant effect on some of the models at the 1 and 5% level. Specifically, in ECA, holding all other variables constant, population growth increased carbon emissions by 0.07%-0.059% in models 1 and 2, respectively. In Model 3, the results indicated that increases in population decreased life expectancy by 0.002% at the 5% significance level.

For MENA, the results showed that population growth produced insignificant results for Model 1. However, for Models 2 and 3, increases in population reduced carbon emissions and life expectancy by 0.04% and 0.07%, respectively. In SSA, increases in population increased carbon emissions by 0.08% and 0.06% for models 1 and 2, respectively. For Model 3, increases in population decreased life expectancy by 0.009%.

Except for SSA in Model 3, where the data showed a negative correlation between GDP per capita and life expectancy, the results showed that GDP per capita had a significant and positive link with carbon emissions and life expectancy. In particular, GDP increased carbon emissions in ECA by 47% to 50% while GDP per capita increased life expectancy by 0.24%, all other factors being held constant. While GDP per capita had no impact on life expectancy in MENA, it increased carbon emissions by 0.63% to 0.76%.

For Models 1 and 2 in SSA, the GDP per person increased carbon emissions by 1.28%. But in Model 3, a rise in GDP per capita resulted in a 0.038% drop in life expectancy. Our findings demonstrate a connection between economic growth and carbon emissions, underscoring the significance of utilizing renewable energy sources. Energy consumption adds to the creation of carbon, and regional differences exist in how population expansion affects emissions. In conclusion, higher per capita GDP is associated with higher carbon emissions, despite the fact that population increase has both positive and negative effects on carbon production and life expectancy. However, in most cases, higher per capita GDP is also linked to higher life expectancy. Tables 8 and 9 presents the results of our findings.

As a post-estimation technique, we used the average marginal effects of all variables to verify the robustness and validity of these findings. The estimated coefficients are shown in Figs 7–18, to be within the 95% confidence interval.

## Discussion of results

The study's findings enhance our understanding of the complex relationships among carbon emissions, life expectancy, GDP per capita, death rate, and population. The research focused on three key regions: ECA, MENA, and SSA. The findings reveal intricate dynamics and interconnections between variables in the studied regions, emphasizing the need for a detailed understanding and exploration of their implications.

Carbon emissions pose a significant health risk to the public, despite global awareness and scientific evidence. Our research emphasizes the ongoing challenge of limiting global warming as carbon emissions persistently exceed desired levels. According to the research, the ECA region has the second-highest average annual carbon emissions (6.88 mt/pc), which is double the emissions of the MENA region. The SSA region has the lowest average annual carbon emissions (0.7449 mt/pc). These findings highlight the critical need for group efforts to reduce carbon emissions and their detrimental effects on public health.

Energy use in the study regions significantly contributes to carbon emissions, indicating a lack of sustainable practices. We determined that a considerable portion of $CO_2$ emissions are as a result of energy use. Overall, 20%-70% of carbon emissions are attributed to energy use. These findings encourage the need to prioritize and implement conservative energy practices to effectively lower carbon emissions in these regions. This finding is Consistent with [42], our findings reveal that carbon emissions vary based on income level. Interestingly, model 3 shows

**Table 8. FGLS estimates of the relationship between Carbon, GDPpcap, Life, POP, and Energy in Sub-Saharan Africa, Middle East and North Africa, and Europe and Central Asia.**

| REGION | ECA | | | MENA | | | SSA | | |
|---|---|---|---|---|---|---|---|---|---|
| MODELS | 1 | 2 | 3 | 1 | 2 | 3 | 1 | 2 | 3 |
| VARIABLES | | | | | | | | | |
| lnLIFE | 1.80*** (8.86) | | | 2.38*** (9.31) | | | 1.33*** (6.75) | | |
| lnCARBON | | | .017*** (8.23) | | | .02*** (3.00) | | | .041*** (7.95) |
| lnDEATH | | .102*** (2.20) | | | -.59*** (-14.72) | | | -.528*** (-7.64) | |
| lnENERGY | .526*** (35.67) | .410*** (22.27) | -.074*** (-39.03) | .74*** (18.28) | .56*** (14.59) | -.051*** (-8.00) | .058*** (1.08) | .046*** (0.88) | -.09*** (-11.58) |
| lnPOP | .032*** (3.47) | .037*** (4.21) | .0006 (0.71) | .031*** (2.28) | -.049*** (-4.00) | -.010*** (-5.52) | .09*** (3.69) | .075*** (3.07) | .003 (0.42) |
| lnGDPpcap | .363*** (25.85) | .408*** (31.01) | .016*** (11.38) | .765*** (35.85) | .60*** (35.21) | .006 (0.99) | 1.28*** (29.44) | 1.27*** (29.65) | -.047*** (-4.92) |
| CONSTANT | -12.382*** (-13.35) | -4.75*** (-19.55) | 4.477*** (193.22) | -19.23*** (-17.39) | -4.56*** (-10.98) | 4.62*** (59.84) | -17.119*** (-14.83) | -10.052*** (-14.55) | 4.77*** (35.37) |
| No OF OBVS./No OF GROUPS | 1,117/47 | 1,121/47 | 1,117/47 | 329/ 15 | 329/15 | 329/ 15 | 487/20 | 487/20 | 487/20 |
| C COMMON AR COEFFICIENT | 0.6048 | 0.5725 | 0.5518 | 0.4241 | 0.4228 | 0.2697 | 0.5469 | 0.5495 | 0.5361 |
| Time dummies | Yes | Yes | Yes | Yes | Yes | Yes | Yes | Yes | Yes |
| Wald Statistic | 1311.62 | 1201.49 | 3686.03 | 5402.95 | 4372.44 | 305.11 | 1420.58 | 1465.91 | 318.86 |

*** = 1% significance

** = 5% significance

* = 10% significance

() = Z-statistics

that energy use negatively affects life expectancy, confirming the link between energy sources and carbon emissions. These results align with [27], who found that energy consumption's environmental impact reduces life expectancy in Pakistan. Based on the results, conservative energy sources must be developed further in order to lessen the detrimental effects of carbon emissions on public health outcomes.

We further reveal that, all models and regional aggregations studied excluding the SSA region in model 3, show a positive association between GDP per capita, carbon emissions, and life expectancy. The results suggest that each increase in GDP per capita is associated with a 0.024% increase in carbon emissions in the ECA and a 0.766% increase in MENA. Our analysis reveals a positive correlation between GDP per capita and carbon emissions, as well as life expectancy, in most regions and models studied. For example, in the ECA region, each increase in GDP per capita is associated with a 0.024% increase in carbon emissions. These findings align with [43] that highlights the role of GDP per capita in driving carbon emissions. Additionally, higher GDP per capita is generally linked to increased life expectancy, as observed in various regions, consistent with [44].

However, the unexpected finding in model 3 for the SSA region, where higher GDP per capita is associated with lower life expectancy, warrants further investigation. This finding may be attributed to environmental deterioration resulting from increased GDP per capita, negatively impacting public health outcomes. It suggests that the benefits of economic prosperity might not have been equitably distributed, leading to increased social and economic

**Table 9. PCSE estimates of the relationship between carbon, GDPpcap, Life, POP, and energy in Sub-Saharan Africa, Middle East and North Africa, and Europe and Central Asia.**

| REGION | ECA | | | MENA | | | SSA | | |
|---|---|---|---|---|---|---|---|---|---|
| MODELS | 1 | 2 | 3 | 1 | 2 | 3 | 1 | 2 | 3 |
| VARIABLES | | | | | | | | | |
| lnLIFE | 1.35*** (2.99) | | | 1.40*** (3.88) | | | 1.43*** (5.35) | | |
| lnCARBON | | | .008*** (2.62) | | | .024*** (2.96) | | | .039*** (5.60) |
| lnDEATH | | .37*** (5.21) | | | -.51*** (-11.10) | | | -.60*** (-6.21) | |
| lnENERGY | .585*** (14.34) | .50*** (15.95) | -.062*** (-20.38) | .72*** (15.77) | .53*** (12.61) | -.05*** (-7.03) | .20*** (3.24) | .209*** (3.28) | -.10*** (-10.14) |
| lnPOP | .07*** (5.01) | .059*** (4.61) | -.002** (-1.89) | .014 (0.87) | -.04*** (-2.95) | -.007*** (-3.58) | .08*** (2.93) | .06*** (2.39) | -.009*** (-2.07) |
| lnGDPpcap | .47*** (18.70) | .50*** (24.05) | .024*** (12.33) | .76*** (29.94) | .63*** (3.75) | .003 (0.62) | 1.28*** (26.71) | 1.28*** (26.27) | -.038*** (-3.43) |
| CONSTANT | -12.47*** (-6.18) | -7.20*** (-16.39) | 4.40*** (142.36) | -14.64 (-8.97) | -4.96*** (-10.28) | 4.58*** (48.32) | -18.12*** (-12.06) | -10.67*** (-12.30) | 4.96*** (34.95) |
| No of obvs/ groups | 1,117/ 47 | 1,121/47 | 1,117/47 | 329/ 15 | 329/15 | 329/ 15 | 487/20 | 487/20 | 487/20 |
| R-SQUARED | 0.64 | 0.65 | 0.99 | 0.92 | 0.9387 | 0.995 | 0.69 | 0.70 | 0.98 |
| Time dummies | Yes | Yes | Yes | Yes | Yes | Yes | Yes | Yes | Yes |
| Wald Statistic | 568.33 | 633.57 | 2397.77 | 3743.25 | 3640.16 | 223.84 | 1353.99 | 1333.66 | 224.73 |

\*\*\* = 1% significance

\*\* = 5% significance

\* = 10% significance

() = Z-statistics

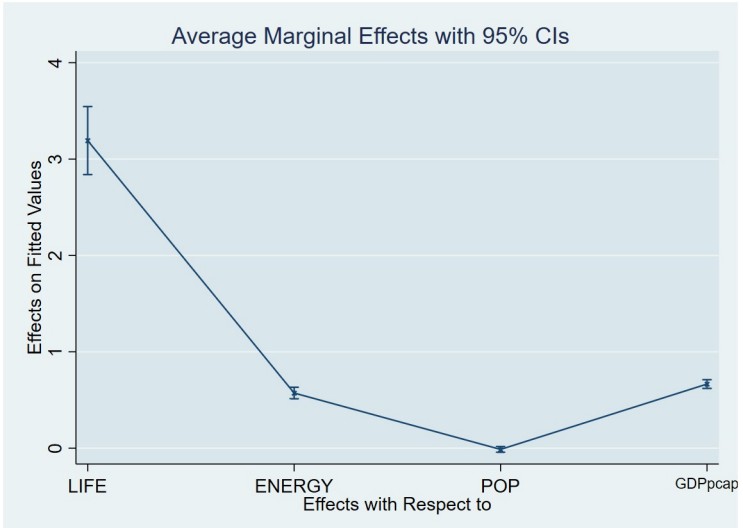

**Fig 7. Margins plot for model 1 illustrating the relationship between carbon emissions, life expectancy, energy, population and GDP per capita in the combined sample.** Marginal plots showing the average marginal effects of all covariates for the combined sample, with a 95% confidence interval. The plot depicts the relationships between carbon emissions and life expectancy, energy, and GDP per capita, as well as the relationships between carbon emissions, death rate, energy, and GDP per capita.

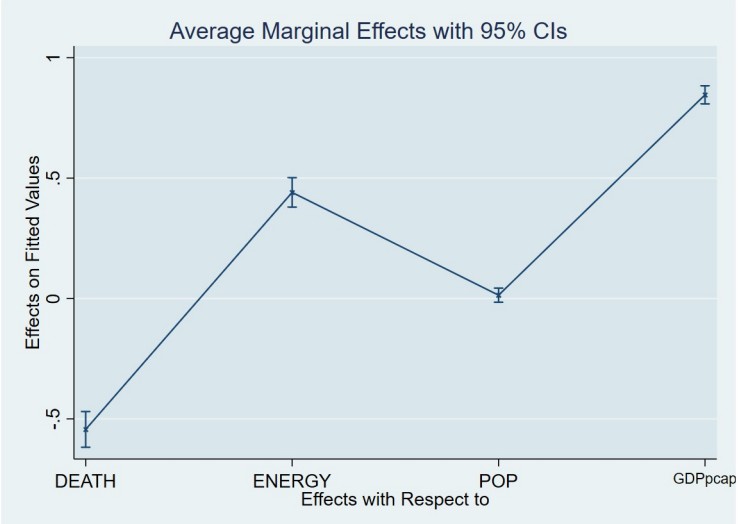

**Fig 8. Margins plot for model 2 illustrating the relationship between carbon, death rate, energy, population and GDP per capita in the combined sample.** Marginal plots showing average marginal effects of all covariates for the combine sample, with a 95% confidence interval. The plot depicts the relationship between carbon emission, death rate, energy, population and GDP per capita.

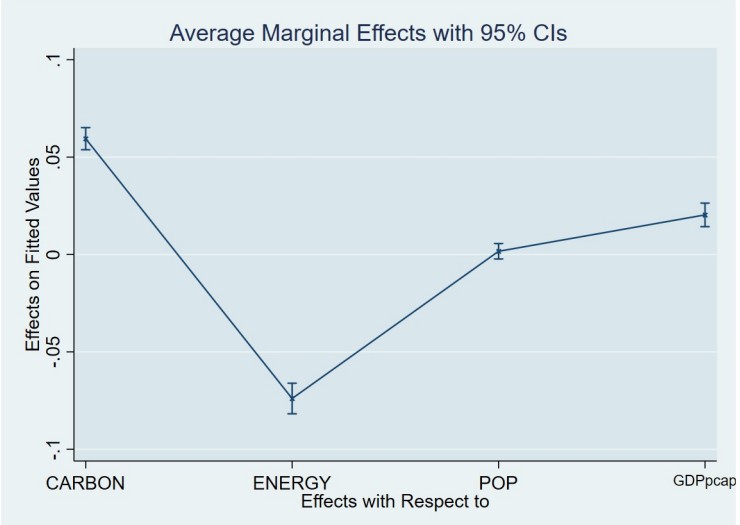

**Fig 9. Margins plot for model 3 illustrating the relationship between life expectancy, carbon emission, energy, population and GDP per capita.** Marginal plots showing average marginal effects of all covariates for the combined sample, with a 95% confidence interval. The plot depicts the relationship between life expectancy, carbon, energy, population, and GDP per capita.

inequality that adversely affects health outcomes. Further research is needed to understand the underlying factors contributing to this phenomenon.

The authors revealed notable variations in death rates and carbon emissions across different geographic regions. Especially in the ECA region, higher carbon emissions are linked to increased death rates, conversely, in the MENA and SSA regions, lower carbon emissions are

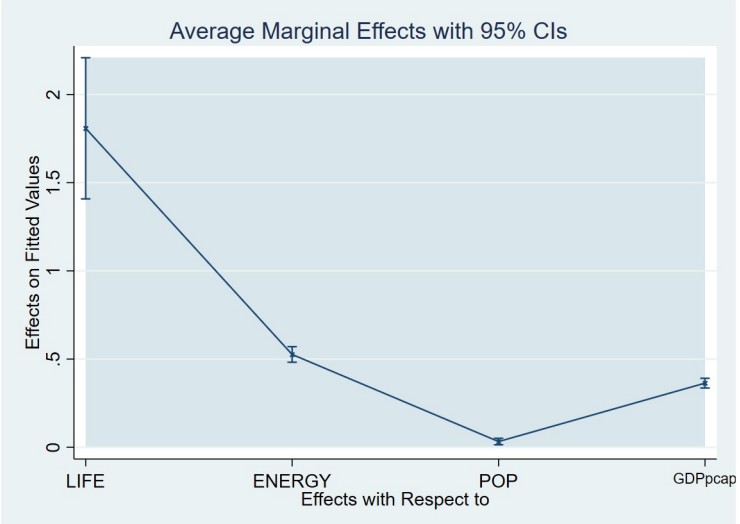

**Fig 10. Margins plot for model 1 showing the relationship between carbon emission, life expectancy, energy, pop, GDP per capita for Europe and Central Asia.** Marginal plots showing average marginal effects of all covariates for Europe and Central Asia, with a 95% confidence interval. The plot depicts the relationship between carbon, life expectancy, energy, population, and GDP per capita.

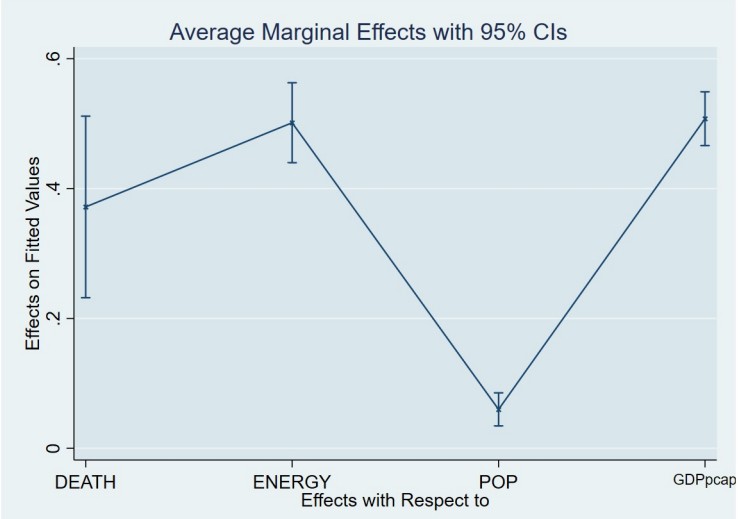

**Fig 11. Margins plot for model 2 showing the relationship between carbon, death rate, energy, population and GDP per capita for Europe and Central Asia.** Marginal plots showing average marginal effects of all covariates for Europe and Central Asia, with a 95% confidence interval. The plot depicts the relationship between carbon, death rate, energy, population, and GDP per capita.

associated with higher death rates. This contrasting relationship suggests that regions with high mortality rates may eventually experience a decrease in population and energy consumption, leading to reduced carbon emissions. Additionally, rising mortality rates may promote environmentally friendly behaviors and the adoption of sustainable practices. Investments in healthcare infrastructure and disease prevention, which improve overall quality of life and

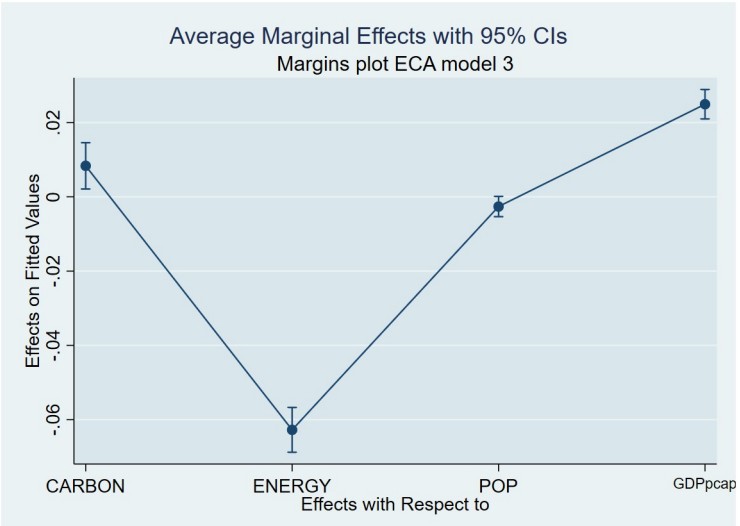

**Fig 12. Margins plot for model 3 showing the relationship between life expectancy, carbon emission, energy, population and GDP per capita for Europe and Central Asia.** Marginal plots showing average marginal effects of all covariates for Europe and Central Asia, with a 95% confidence interval. The plot depicts the relationship between carbon, life expectancy, energy, population, and GDP per capita.

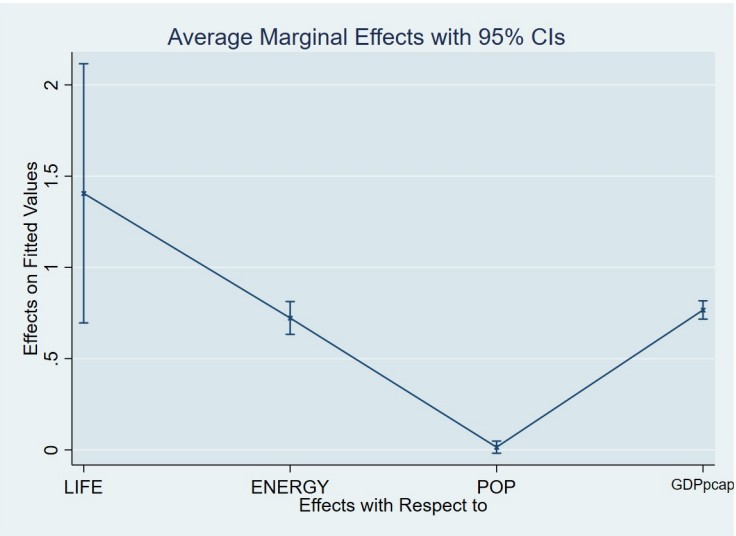

**Fig 13. Margins plot for model 1 showing the relationship between carbon emission, life expectancy, energy, population, GDP per capita for Middle-East and North Africa.** Marginal plots showing average marginal effects of all covariates for Middle-East and North Africa, with a 95% confidence interval. The plot depicts the relationship between carbon, life expectancy, energy, population, and GDP per capita.

reduce energy-intensive medical treatments, can indirectly contribute to lower carbon emissions. These findings highlight the complex and context-dependent nature of the link between death rates and carbon emissions.

Although on a very modest scale, our research's model 3 results show that carbon emissions have an impact on life expectancy. We discovered that life expectancy rises along with carbon

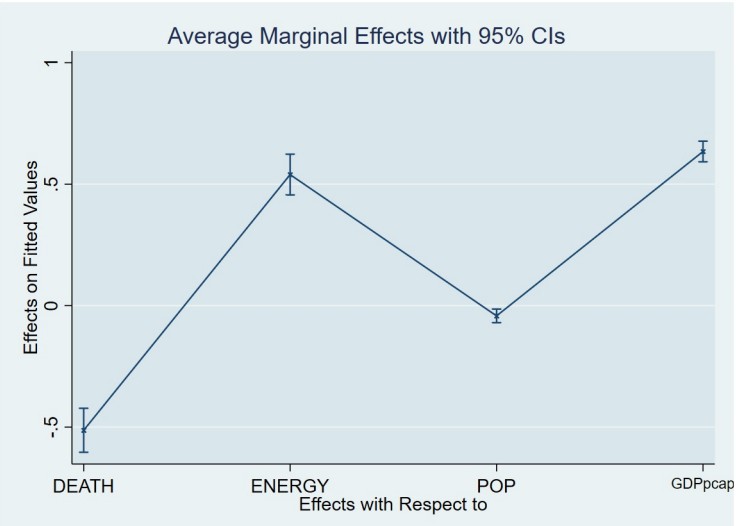

**Fig 14. Margins plot for model 2 showing the relationship between carbon emission, death rate, energy, population and GDP per capita for Middle-East and North Africa.** Marginal plots showing average marginal effects of all covariates for Middle-East and North Africa, with a 95% confidence interval. The plot depicts the relationship between carbon, death rate, energy, population, and GDP per capita.

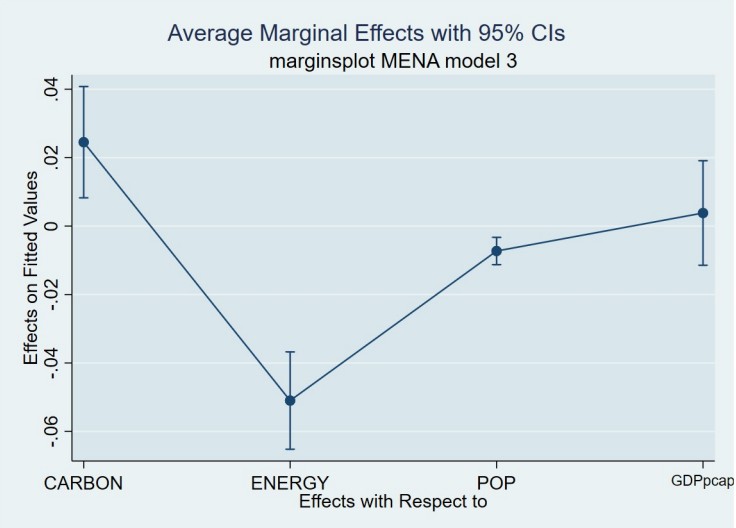

**Fig 15. Margins plot for model 3 showing the relationship between life expectancy, carbon emission, energy, population and GDP per capita for Middle-East and North Africa.** Marginal plots showing average marginal effects of all covariates for Middle-East and North Africa, with a 95% confidence interval. The plot depicts the relationship between life expectancy, carbon emission, energy, population, and GDP per capita.

emissions, increasing by 0.008% to 0.039% for every percentage increase in carbon emissions. These results are consistent with [23] who discovered that carbon emissions in poor nations increase life expectancy. They suggested that consumption, not production, may be to blame for this incidence. Our study suggests that the positive association between carbon emissions and life expectancy may be explained by improved economic activity. Higher carbon

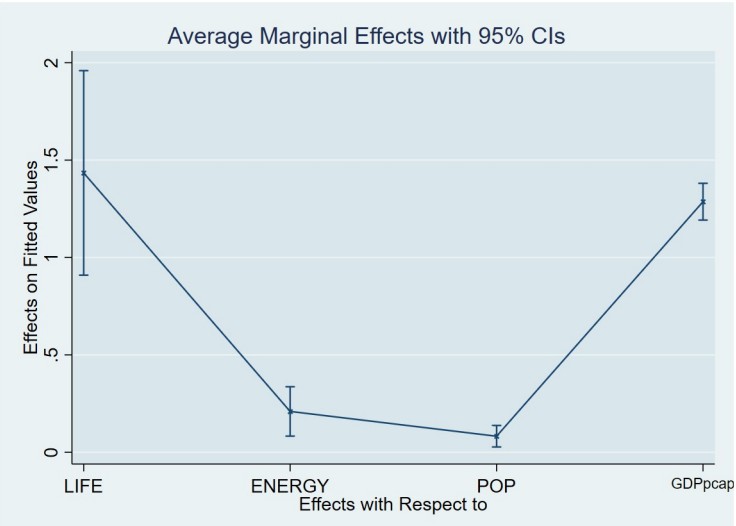

**Fig 16. Margins plot for model 1 showing the relationship between carbon emission, life expectancy, energy, pop, GDP per capita for Sub-Saharan Africa.** Marginal plots showing average marginal effects of all covariates for Sub-Saharan Africa, with a 95% confidence interval. The plot depicts the relationship between carbon, life expectancy, energy, population, and GDP per capita.

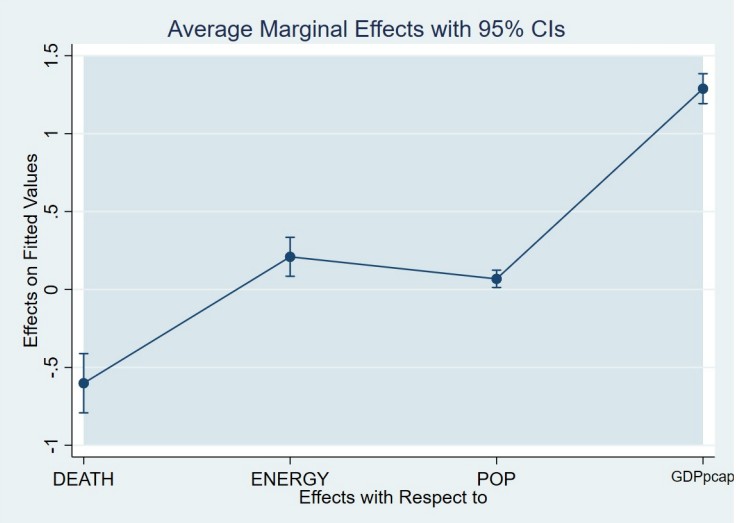

**Fig 17. Margins plot for model 2 showing the relationship between carbon emission, death rate, energy, population and GDP per capita for Sub-Saharan Africa.** Marginal plots showing average marginal effects of all covariates for Sub-Saharan Africa, with a 95% confidence interval. The plot depicts the relationship between carbon, death rate, energy, population, and GDP per capita.

emissions, often associated with industrialization, can lead to job creation and economic growth, which in turn contribute to better healthcare access, nutrition, and overall resources supporting good health. Industrialization is commonly linked to carbon emissions and has the potential to positively impact public health.

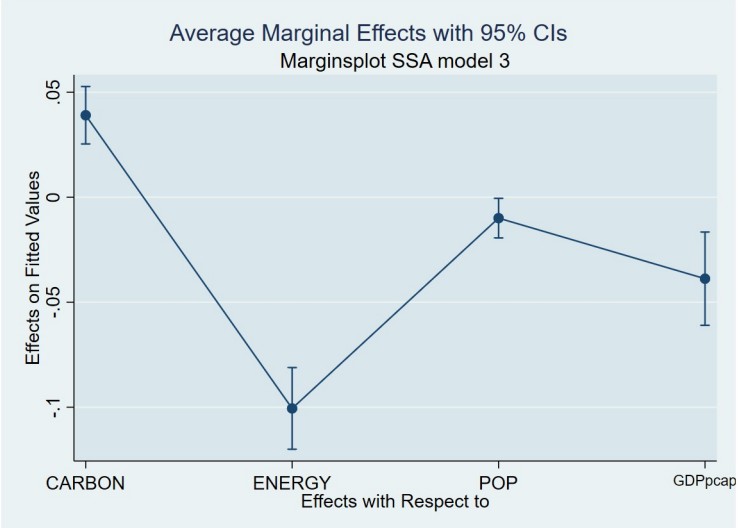

**Fig 18. Margins plot for model 3 showing the relationship between life expectancy carbon emission, energy, population and GDP per capita for Sub-Saharan Africa.** Marginal plots showing average marginal effects of all covariates for Sub-Saharan Africa, with a 95% confidence interval. The plot depicts the relationship between life expectancy, carbon emission, energy, population, and GDP per capita.

Our research shows a positive correlation between population increase and carbon emissions in models 1 and 2 for ECA and SSA regions, indicating that carbon emissions rise with population growth in these areas. Additionally, model 3 reveals a negative correlation between population growth and life expectancy across all regions studied. This highlights the importance of implementing sustainable policies to address the challenges associated with population growth. Interestingly, in model 2 for MENA, carbon emissions decrease as population grows, suggesting a potentially unique relationship influenced by specific circumstances.

## Further analysis

Our research sought to understand the nature of the association between carbon emissions and GDP per capita and to determine whether there are regional differences in this association. Regression analysis was done on a sample of nations from the SSA, MENA, and ECA areas in order to achieve this. Our analysis showed that the connection between carbon emissions and GDP per capita in the combined sample is inverted U-shaped. When GDP per capita was regressed on carbon emissions, we found a statistically significant negative coefficient on the cubic term. The Environmental Kuznets Curve (EKC) hypothesis, which contends that environmental deterioration first rises with economic growth but subsequently falls as nations get wealthy, is supported by this study [4].

The initial model in SSA yielded negligible results. However, the cubic factor, which was added to model 5 was shown to be significant and negative, suggesting an N-trajectory link between the region's carbon emissions and GDP per capita. A non-monotonic relationship between two variables—one where the relationship changes direction at least once—is referred to as an N-trajectory relationship. The N-trajectory association between economic expansion and environmental deterioration has been observed in earlier studies, and this finding is consistent with those studies [7]. To the best of our knowledge this study is the first to find the N-trajectory in sub-Saharan Africa. Both tests in MENA were not statistically significant.

**Table 10. Regression results for the Environmental Kuznets Curve: Quadratic and cubic analysis of the relationship between carbon emissions and GDP per capita.**

| SAMPLES | TOTAL SAMPLE | | SSA | | MENA | | ECA | |
|---|---|---|---|---|---|---|---|---|
| MODELS | 4 | 5 | 4 | 5 | 4 | 5 | 4 | 5 |
| VARIABLES | | | | | | | | |
| LnCARBON | | | | | | | | |
| LnLIFE | 2.94*** (28.00) | N/A | 1.358 (4.67) | 1.50 (5.09) | 1.31 (3.67) | 1.357 (3.76) | 1.49 (3.48) | N/A |
| LnENERGY | .61*** (33.28) | N/A | .217 (2.99) | .22 (3.11) | .726 (15.90) | .726 (16.36) | .56 (13.63) | N/A |
| LnGDPpcap | 3.67*** (24.94) | N/A | 1.04 (1.83) | -16.03 (-2.88) | .763 (2.26) | -6.97 (-1.69) | 2.2 (7.30) | N/A |
| Lngdppcap$^2$ | -.17 (-16.41) | N/A | .0135 (0.36) | 2.36 (3.06) | -.0003555 (0.985) | .839 (1.89) | -.096 (-5.93) | N/A |
| Lngdppcap$^{3.}$ | N/A | N/A | N/A | -.106 (-3.02) | N/A | -.03 (-1.90) | N/A | N/A |
| N | 1,933 | N/A | 487 | 487 | 329 | 329 | 1,117 | N/A |
| CONSTANT | -33.38 (-37.15) | N/A | -15.46154 (-5.19) | 24.79909 (1.90) | -13.98038 (-6.31) | 9.41493 (0.76) | -19.56 (-7.77) | N/A |
| *KUZNETS CURVE* | | | | | | | | |
| PRESENCE OF INVERSE U-SHAPE | YES | N/A | NO | NO | NO | NO | YES | N/A |
| INVERTED N-SHAPE | NO | N/A | N/A | YES | N/A | YES | NO | N/A |

$H_1$: Inverse U shape

$H_0$: Monotone or U shape

Lngdppcap$^2$ = square root of per capita GDP

Lngdppcap$^3$ = cubic root of per capita GDP

**Table 11. U-Test results for the relationship between carbon emissions and GDP per capita: Combined sample and Europe and Central Asia.**

| samples | Total sample | | Europe and central Asia | |
|---|---|---|---|---|
| | Lower bound | Upper bound | Lower bound | Upper bound |
| Interval | 5.248256 | 11.62998 | 5.248256 | 11.62998 |
| Slope | 1.924041 | -.3150482 | 1.20298 | -.0248444 |
| t-value | 25.65108 | -5.837296 | 8.978437 | -.3139682 |
| P > \|t\| | 1.9e-125 | 3.10e-09 | 5.70e-19 | .3768019 |
| Extreme point | 10.73205 | | 11.50085 | |
| Overall test of the presence of an inverse U-shape | | | | |
| t-value | 5.84 | | 0.31 | |
| P > \|t\| | 3.10e-09 | | .377 | |
| specification | f(x) = x^2 | | f(x) = x^2 | |

$H_1$: Inverse U shape

$H_0$: Monotone or U shape

This implies that there might not be a direct link between carbon emissions and economic growth in this area or that other variables besides GDP per capita might be more significant in explaining carbon emissions. Finally, we discovered evidence of an inverted U-shaped link between economic growth and carbon emissions in ECA. This result is in line with earlier studies that showed an EKC link in this area [45]. We tested for the validity of the models using the utest [46]. The results of the environmental Kuznets curve and the utest are presented in Tables 10 and 11.

## Conclusion and policy implications

In this study, three diverse regions ECA, the MENA, and SSA are examined to better understand the complex interactions between many variables and their effects on carbon emissions

and life expectancy. Five distinct models are used in the study; three of them look for linear links between the variables, while the other two look for non-linear interactions between carbon emissions and GDP per capita. The study does a regional analysis as well in order to present important findings and provide insightful policy recommendations.

The study's conclusions offer important new understandings of the connections between utilized variables, their effects on carbon emissions and life expectancy in the ECA, MENA, and SSA areas, and their importance to meeting the Sustainable Development Goals (SDGs). By implementing these measures, we can move closer to achieving the SDG 7, which aims to ensure access to affordable, reliable, sustainable, and modern energy for all.

Public health initiatives: Considering that the results from Model 3 have demonstrated that increasing energy consumption is associated with a decrease in life expectancy, governments should prioritize the establishment of public health initiatives to mitigate the adverse effects of rising energy consumption. In doing so, governments would actively contribute to the realization of SDG 3, which aims to "ensure healthy lives and promote well-being for all ages." The correlation between GDP per capita and carbon emissions is positive, highlighting the urgency for implementing sustainable economic development measures. Policymakers should give precedence to promoting economic growth decoupled from carbon emissions. This can be accomplished by advocating sustainable corporate practices, investing in green technologies, and fostering innovation in clean industries. A well-executed approach in this direction could bring us closer to the attainment of SDG 1 and 10, which calls for the "eradication of poverty which is linked to income levels" and "reduction of inequality within and among countries".

Regional context, regional disparities found in the study, and the significance of context-specific strategies are all highlighted. Interventions should be tailored by policymakers to the distinctive qualities of each location, taking into account elements including the economy, demographic trends, and environmental issues. This may entail working with regional stakeholders, performing additional research, and using a multifaceted strategy to address particular area needs.

Integrated strategy: Policymakers should adopt an integrated approach that takes into account the linkages between energy, population, economics, and health given the interplay of multiple aspects that was identified in the research. This may entail coordinating initiatives amongst many sectors and creating comprehensive strategies that simultaneously address various elements. This strategy might produce results that are more enduring and efficient. In addition to the foregoing, the EKC analysis offers policy makers crucial implications for promoting sustainable development and reducing the adverse environmental effects of economic expansion.

First off, our findings imply that the EKC hypothesis is accurate for the total sample of countries, showing that richer nations tend to adopt greener practices and technologies. By encouraging policies that support energy-efficient living, encourage the use of green technologies, and invest in renewable energy sources, policymakers can take advantage of this understanding.

Second, the N-trajectory found in SSA necessitates customized policies that considers the unique environmental and economic issues in this region. Policymakers in SSA may need to prioritize fostering economic growth while also tackling environmental deterioration, such as through encouraging reforestation initiatives, investing in sustainable agricultural techniques, and enhancing waste management systems.

Thirdly, the lack of a direct correlation between MENA carbon emissions and economic growth raises the possibility that other variables other than GDP per capita may be more crucial in explaining MENA carbon emissions. Political instability, social unrest, and violence are just a few of the other factors that policymakers in the MENA region may need to concentrate on in order to stop environmental deterioration.

Not to mention, the EKC association discovered in ECA suggests that decision-makers in this location can leverage economic growth to help sustainable development. They will need to implement policies like promoting renewable energy sources, improving energy efficiency, and rewarding the usage of green technologies in order to encourage the adoption of environmentally friendly behaviors and technologies.

Overall, our study emphasizes the significance of supporting sustainable development in a nuanced manner that takes into account the distinct economic and environmental issues that various parts of the world face. When formulating policies aiming at reducing the damaging effects of economic expansion on the environment, policymakers and stakeholders should take the lessons learned from our study into account.

## Supporting information

**S1 File. Countrynames names of countries in the sample.**
(XLSX)

**S1 Table. Dickey-fuller unit root test.**
(DOCX)

## Acknowledgments

We express our gratitude to Prof. Cisheng Wu for his invaluable supervision and guidance, which enabled us to successfully complete this project. We also extend our appreciation to miss Teng Liu for proofreading and providing helpful suggestions. Lastly, we are thankful to the management school of Hefei University of Technology for providing us with the opportunity to carry out this project.

## Author Contributions

**Conceptualization:** Frank Osei-Kusi.

**Data curation:** Frank Osei-Kusi, Stephen Tetteh, Wendy Irena Guerra Castillo.

**Formal analysis:** Frank Osei-Kusi, Stephen Tetteh.

**Funding acquisition:** Cisheng Wu.

**Investigation:** Frank Osei-Kusi.

**Methodology:** Frank Osei-Kusi.

**Project administration:** Frank Osei-Kusi.

**Resources:** Frank Osei-Kusi, Wendy Irena Guerra Castillo.

**Software:** Frank Osei-Kusi.

**Supervision:** Cisheng Wu.

**Validation:** Frank Osei-Kusi, Cisheng Wu.

**Visualization:** Frank Osei-Kusi.

**Writing – original draft:** Frank Osei-Kusi.

**Writing – review & editing:** Frank Osei-Kusi, Cisheng Wu, Stephen Tetteh, Wendy Irena Guerra Castillo.

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
