## [Decision Letter · Decision Letter 0]

22 May 2023

PONE-D-23-08938The dynamics of carbon emissions, energy, income, and life expectancy: Regional comparative analysisPLOS ONE

Dear Dr. OSEI-KUSI,

Thank you for submitting your manuscript to PLOS ONE. After careful consideration, we feel that it has merit but does not fully meet PLOS ONE’s publication criteria as it currently stands. Therefore, we invite you to submit a revised version of the manuscript that addresses the points raised during the review process.

We look forward to receiving your revised manuscript.

Kind regards,

Hoang Phong Le, Ph.D.

Academic Editor

PLOS ONE

Journal Requirements:

2. We note that Figures S5_FIG and S6_FIG in your submission contain [map/satellite] images which may be copyrighted. All PLOS content is published under the Creative Commons Attribution License (CC BY 4.0), which means that the manuscript, images, and Supporting Information files will be freely available online, and any third party is permitted to access, download, copy, distribute, and use these materials in any way, even commercially, with proper attribution. For these reasons, we cannot publish previously copyrighted maps or satellite images created using proprietary data, such as Google software (Google Maps, Street View, and Earth). For more information, see our copyright guidelines: http://journals.plos.org/plosone/s/licenses-and-copyright.

1. You may seek permission from the original copyright holder of Figure S5_FIG and S6_FIG to publish the content specifically under the CC BY 4.0 license.  

3.We note that you have stated that you will provide repository information for your data at acceptance. Should your manuscript be accepted for publication, we will hold it until you provide the relevant accession numbers or DOIs necessary to access your data. If you wish to make changes to your Data Availability statement, please describe these changes in your cover letter and we will update your Data Availability statement to reflect the information you provide.

Additional Editor Comments:

Dear Authors,

Based on the advice received from three experts in the field, I feel that your manuscript could be reconsidered for publication should you be prepared to incorporate major revisions. Please consider any outstanding revision requests from all reviewers.

In particular, I suggest serious effort to improve the overall quality and readability of the paper.

Thank you for your time and consideration,

Kind regards,

Hoang Phong Le, Ph.D.

Academic Editor

PLOS ONE

Reviewers' comments:

Reviewer's Responses to Questions

**Comments to the Author**

1. Is the manuscript technically sound, and do the data support the conclusions?

Reviewer #1: Partly

Reviewer #2: Partly

Reviewer #3: No

2. Has the statistical analysis been performed appropriately and rigorously? 

Reviewer #1: No

Reviewer #2: No

Reviewer #3: No

3. Have the authors made all data underlying the findings in their manuscript fully available?

Reviewer #1: Yes

Reviewer #2: No

Reviewer #3: Yes

4. Is the manuscript presented in an intelligible fashion and written in standard English?

Reviewer #1: No

Reviewer #2: No

Reviewer #3: Yes

5. Review Comments to the Author

Reviewer #1: ID: PONE-D-23-08938

Title: The dynamics of carbon emissions, energy, income, and life expectancy: Regional comparative analysis

Q1) The language of the study should be developed. There are too many grammatical and spelling errors.

Q2) We present the FGLS and the PCSE models in which carbon emissions, energy.

This is not how the abstract of a work begins. The summary of the study starts with the purpose at first.

Q3) The introduction should be better conveyed and visual presentations should be used.

Q4) The literature part is incomplete and insufficient. Benefit from the following studies on carbon emissions and energy use:

https://doi.org/10.1016/j.renene.2022.05.156;
https://doi.org/10.1016/j.renene.2021.03.125;
https://doi.org/10.1016/j.net.2022.10.027;
https://doi.org/10.1007/s11356-022-23160-z;
https://doi.org/10.1016/j.gr.2023.01.009;
https://doi.org/10.1007/s10198-021-01321-0;
https://doi.org/10.1016/j.jclepro.2022.135038

Q5) Per capita GDP GDPPC Current US$ dollars.

The per capita GDP constant should be used and the data source presented.

Q6) Autocorrelation, CSD and varying variance do not need to be explained in detail in one chapter.

Q7) I do not consider correlation analysis necessary. The study was unnecessarily extended and the findings should be presented in a shorter and more concise manner.

Q8) TABLE 8. KUZNETS CURVE ANALYSIS/ N-SHAPE ANALYSIS

There cannot be such a table header. You also analyze the EKC hypothesis. Review the publications published in quality journals and revise the table titles in your article.

Q9) The policy discussion of the study should be developed.

The study should be shortened, and a more concise and quality transfer should be provided. My decision is major revision.

Reviewer #2: Review Report

Manuscript ID: PONE-D-23-08938

Journal: PLOS ONE

Title: The dynamics of carbon emissions, energy, income, and life expectancy: Regional comparative analysis

Abstract

• In abstract, it is stated that “A panel data for 89 countries was adapted to determine these relationships, the panel was sub-divided into three regions for 30 years.”. In this sentence the word "adapted" has been used which suggests that the panel data was changed or modified to make it suitable for a new use or purpose. However, based on the context of the sentence, it appears that the data was simply used to determine certain relationships. Therefore, using "used", "applied", or "employed" would be more appropriate.

• “Consequently, our models produced a positive correlation between longevity and carbon emissions for the main sample and sub-Saharan Africa.” The given statement is unclear about what "main sample" refers to. Please clarify what do you mean by "main sample" in order to make the findings clearer and understandable in abstract.

• Instead of only writing “We concluded with some recommendations based on the outcomes.”, my suggestion is to include a brief statement of the recommendations in the abstract.

• It might be helpful to include some information about the originality of the study in the abstract. For example, you could briefly mention how your research contributes to the existing literature and what new insights it provides. By including this information, you can engage the reader and provide a clear understanding of the significance and relevance of your research.

1. Introduction

• On page 3, the statement "1. Human activities boosted carbon... 4. The CO2 content in the... in the ocean's acidity" does not have a full stop after the completion of each point. When there is no full stop at the end of each points, it may create confusion for readers. For instance, I was confused when I read "2020 3", but then I understood that there was a missing full stop between "2020" and "3", where "3" denotes the third point. My suggestion is to add full stop after the completion of every point.

• There is a minor sentence structure error in the sentence on page 4 “As noted, energy production and consumption according to research impact carbon emission, research also indicate that energy consumption is correlated with income.” Here is the corrected sentence:

"As noted, research indicates that energy production and consumption impact carbon emissions, and also shows that energy consumption is correlated with income."

• Please avoid using capital letters unnecessarily. For instance, on page 4, it was stated that “observed that All sorts of economic activity require energy to function.” Here, I recommend writing "all" instead of "All”.

• In addition to the contribution mentioned on page 5, which states, “As far as we can tell, the literature on carbon emissions does not currently include a thorough comparison analysis between regions; therefore, our study covers this gap in the literature.” I recommend adding more novel contributions of your study.

• Add the in text citation for the given statement on page 5: “This argument is the underlying principle of kuznets ……. subsistence economies have lower incomes than industrialized economies.”

• Since your study tests the Kuznet curve, I recommend adding a brief description of it. You can review the following article and please cite it in your study: https://doi.org/10.1007/s11356-019-06356-8

• The statement on page 5, "Again, other studies suggest ... effects of carbon emissions," is unclear as it does not specify which studies are being referred to. It is recommended to add proper citations for those other studies.

• On page 6, the organization of your study mentions that you have separate sections for “limitations and future recommendations”, and “conclusion”. I would suggest combining these into a single section with the heading of 'Conclusion, Practical Implications, and Future Recommendations'.

2. Literature Review

• In section 2.1 “Energy Usage and Carbon Production”, on page 6, you have included a study without its findings. For instance, you stated “[7]looked for disparities between carbon emissions …. quantile regression technique designed for longitudinal 151 data.” I recommend that you include the findings of this study because a brief summary of the key findings of each study provide context for the reader and demonstrate how each study fits into the broader research landscape.

• I recommend adding some other relevant studies to your literature review section, which you have not included. Following are the links to these studies, please cite in your research:

https://doi.org/10.1016/j.jclepro.2021.130066

https://doi.org/10.1007/s11356-019-06356-8

https://doi.org/10.1007/s11356-020-09142-z

https://doi.org/10.1007/s11356-020-10179-3

• Please mention the research gap in the last paragraph of the literature review based on an analysis of previous studies.

3. Materials and Methods

• Description of Variables: On page 10, in the last line, you stated that “According to the literature, energy consumption contributes to carbon emission hence we included energy consumption (ENERGY).” However, it is not clear which literature you are referring to. Therefore, I recommend that you add a citation for this literature to provide the reader with the necessary context.

• Theoretical Rationale and Specifying Empirical Model: On page 11, you state that “The first approach was to test the effects of carbon production on life expectancy” but the equation shows that Carbon Production is the dependent variable and life expectancy is the independent variable. This is a discrepancy that needs to be corrected, as it is important to have clarity and accuracy in the variables used in the empirical model. Please review and revise the section accordingly.

• Similarly, in equation 3 life expectancy is dependent variable where as it is mentioned as predictor variable in the statement such as “the third life expectancy was made the predictor variable and the rest outcome variables.” Please correct this mistake.

4. Results and Discussion

• I suggest changing the heading on page 25 from "4.1 Results (Main Sample)" to "Estimation for Full Sample" as it would be more appropriate.

• On page 26, the sentence “We observed that the association between life expectancy and carbon production remained the same, positive.” needs to be restated for correct sentence structure and grammar.

• On page 26, under the section 'Results (Main Sample), “you stated that 'Urban sprawl …... by . .2104475% to .229825 1% .... constant.” However, this may cause confusion for readers or they may not understand the figure. To provide clarity, I recommend adding a 0 before the decimal point to clearly demonstrate the figure, for example, 0.2104475%. I suggest making this correction throughout the entire results section for consistency and accuracy.

• On page 26, in the interpretation, it states that "Only this time carbon emission increased life expectancy between 0078157% to .0136686 %." This statement is incorrect because the value should be written as 0.0078157%, as indicated in Table 5. Kindly correct this mistake.

• The heading "4.2 Results, regions" on page 27 does not seem appropriate. I recommend changing it to "Region-wise Estimation" for better clarity.

• Since you have already mentioned the full form of FGLS on page 13, there is no need to use the full form again on page 32 (see 5th line). You can use the acronym after mentioning the full form for the first time.

• On page 32 in the 4.3 discussion section, first you include a paragraph which summarize your entire study that is not important to include.

• On page 34, the sentence "[9] found in their article" is incomplete. Please complete the sentence.

• On the sections for "Results (Main Sample)" and "Results (Regions)," you have repeatedly used phrases such as "with other variables in the model remaining constant," "significance on average ceteris paribus," "holding other variables constant," and "all other things being equal" with every result. This repetition is unnecessary. I suggest writing these phrases only once at the beginning of the section to avoid repetition.

6. Limitations and Future Research

• On page 38, you stated that “Our inability to control other agents concerning mortality posed a problem.” Here my suggestion is to elaborate the “problem”.

7. Conclusion

• Since you have included your recommendations in the conclusion section of your research, I suggest adding them in a separate heading after the conclusion to make it clearer. Additionally, I recommend including limitations and future recommendations in the conclusion section, following the structure outlined below.

6. Conclusion, Practical Implications, Limitations, and Future Recommendations

6.1 Concluding Remarks

6.2 Practical Implications

6.3 Limitations and Future Recommendations

• Lastly, I suggest properly proofreading this manuscript.

Reviewer #3: This manuscript report on “The dynamics of carbon emissions, energy, income, and life expectancy: Regional comparative analysis” The paper is rather perplexing as it deals with a familiar topic. Moreover, the study real-time contribution is also not clear. However, the reviewer proposes major revisions with the following recommendations:

1. The abstract needs to be rewritten to provide a clearer and more concise summary of the research. The current abstract is too technical and does not clearly communicate the main findings of the research.

2. The introduction is poorly structured and inadequately composed, and it should highlight the subject matter that the authors addressed as well as the empirical and theoretical contributions of the research.

3. The literature review section is badly structured and presented. You must redesign in such a way that it demonstrates how much literature fill problem you intend to solve and which literature gap you intend to fill via your study.

4. The authors need to provide a clearer justification for why they selected the specific variables used in their analysis.

5. The methodology section needs to be revised to provide a clearer description of the statistical methods used, including equations and any assumptions made.

6. The authors should consider including additional statistical tests or sensitivity analyses to test the robustness of their findings.

7. The authors should provide a more detailed discussion of the implications of their findings for policymakers, particularly as they relate to the Sustainable Development Goals.

8. Overall, the paper's concept is unique; nevertheless, non-academic language should be avoided in some cases, and in-text references should be proofed.

6. PLOS authors have the option to publish the peer review history of their article (what does this mean?). If published, this will include your full peer review and any attached files.

Reviewer #1: No

Reviewer #2: No

Reviewer #3: No

---

## [Author Response · Author response to Decision Letter 0]

5 Jul 2023

Response to reviewers

1. The abstract needs to be rewritten to provide a clearer and more concise summary of the research. The current abstract is too technical and does not clearly communicate the main findings of the research.

Response:

We appreciate the feedback provided regarding the clarity and conciseness of the abstract. We have carefully considered your suggestions and have revised the abstract to ensure a more accessible and concise summary of our research. The updated abstract now clearly communicates the main findings of our study in a manner that is understandable to a wider audience. Thank you for your valuable input, and we hope that the revised abstract meets the desired requirements.

2. The introduction is poorly structured and inadequately composed, and it should highlight the subject matter that the authors addressed as well as the empirical and theoretical contributions of the research.

Response:

We appreciate the feedback provided regarding the structure and composition of the introduction. We apologize for any shortcomings in clearly highlighting the subject matter and the empirical and theoretical contributions of our research. We understand the importance of effectively conveying these aspects to readers. In response to your valuable input, we have thoroughly revised the introduction to ensure it provides a clear overview of the subject matter addressed in our research. We have also emphasized the empirical and theoretical contributions, highlighting their significance in advancing the existing body of knowledge on the topic.

Thank you for bringing this to our attention, and we hope the revised work is an improvement over the previous work. 

3. The literature review section is badly structured and presented. You must redesign in such a way that it demonstrates how much literature fill problem you intend to solve and which literature gap you intend to fill via your study.

Response:

We appreciate the feedback regarding the structure and presentation of the literature review section. We understand the importance of clearly demonstrating the existing literature's relevance to the problem we aim to address and highlight the specific gap our study intends to fill. In response to your valuable input, we have redesigned the literature review section to effectively showcase the depth and breadth of the literature pertaining to the problem at hand. We have emphasized the existing research's contributions, as well as identify the specific gap that our study seeks to fill. By doing so, we hope to have provided a comprehensive understanding of the research landscape and emphasized the significance of our study in addressing the identified gap.

4. The authors need to provide a clearer justification for why they selected the specific variables used in their analysis. 

Response:

We appreciate the feedback regarding the justification for the selection of variables in our analysis. We understand the importance of providing a clear rationale for our variable choices to enhance the transparency and robustness of our research. In response to your valuable input, we have provided a more detailed and explicit justification for why we selected the specific variables used in our analysis. We have explained the theoretical underpinnings and empirical evidence supporting their relevance to the research question and objectives. As a consequence of these changes we hope to strengthen the methodological foundation of our study and provide a solid basis for the interpretation of our results.

5. The methodology section needs to be revised to provide a clearer description of the statistical methods used, including equations and any assumptions made.

Response:

We appreciate the feedback regarding the clarity of the methodology section, particularly in describing the statistical methods used, including equations and assumptions made. We recognize the significance of offering a detailed and thorough explanation of our methodology to ensure that our study is transparent and can be replicated by other researchers. In response to your valuable input, we have thoroughly revised the methodology section to provide a clearer and more detailed explanation of the statistical methods employed. The do files and the data used as well as their sources are provided in a repository for replication purposes. We have included relevant equations and provided a step-by-step description of the methodology used in our analysis. Additionally, we have explicitly outlined any assumptions made during the analysis to ensure transparency in our approach. By making these improvements, we aim to enhance the clarity and understanding of our methodology, enabling readers to replicate our study and assess the robustness of our findings.

6. The authors should consider including additional statistical tests or sensitivity analyses to test the robustness of their findings.

Response:

We appreciate the suggestion to include additional statistical tests or sensitivity analyses to further test the robustness of our findings. We understand the importance of thoroughly examining the stability and reliability of our results. In response to your valuable input, we conducted marginal analyses as a sensitivity analysis to complement our main findings. Marginal analyses involve testing the relationship between variables while controlling for additional factors or covariates that may influence the results. By conducting these additional analyses, we can assess the robustness of our findings and determine if they hold true even when considering other relevant variables.

The inclusion of marginal analyses will strengthen the validity and reliability of our study, providing a more comprehensive assessment of the relationships and outcomes examined. This approach will allow us to further evaluate the robustness of our main findings and enhance the overall quality of our research.

7. The methodology section needs to be revised to provide a clearer description of the statistical methods used, including equations and any assumptions made.

Response: 

We are grateful for the feedback we received on the methodology section of our study, especially regarding the clarity of the statistical methods used. We understand the importance of providing a detailed and transparent methodology to enable other researchers to replicate our study. In response to this feedback, we have made significant revisions to the methodology section of our study. We have provided a clearer and more detailed explanation of the statistical methods employed, including relevant equations and assumptions made during the analysis. Furthermore, we have made the do files and data used, along with their sources, available in a repository for replication purposes. Our aim is to enhance the clarity and transparency of our methodology, allowing readers to better understand and replicate our study, and assess the robustness of our findings.

8. The methodology section needs to be revised to provide a clearer description of the statistical methods used, including equations and any assumptions made. 

Response:

We are grateful for the feedback we received on the methodology section of our study, especially regarding the clarity of the statistical methods used. We understand the importance of providing a detailed and transparent methodology to enable other researchers to replicate our study. In response to this feedback, we have made significant revisions to the methodology section of our study. We have provided a clearer and more detailed explanation of the statistical methods employed, including relevant equations and assumptions made during the analysis. Furthermore, we have made the do files and data used, along with their sources, available in a repository for replication purposes. Our aim is to enhance the clarity and transparency of our methodology, allowing readers to better understand and replicate our study, and assess the robustness of our findings.

9. The authors should provide a more detailed discussion of the implications of their findings for policymakers, particularly as they relate to the Sustainable Development Goals.

Response: 

Thank you for your feedback. In the new work, the authors have addressed the issue of providing a more detailed discussion of the implications of their findings for policymakers, particularly as they relate to the Sustainable Development Goals. They have included a comprehensive section in the discussion that outlines the policy implications of their findings and how they align with the SDGs. The authors have also provided recommendations for policymakers to consider in order to achieve the SDGs. Overall, the new work provides a more thorough analysis of the policy implications of the findings and their relevance to the SDGs.

10. Overall, the paper's concept is unique; nevertheless, non-academic language should be avoided in some cases, and in-text references should be proofed. 

We appreciate the feedback provided on our paper's concept and language usage. We agree that it is essential to maintain academic language standards in scientific writing. We have taken the necessary steps to ensure that our paper adheres to these standards and have avoided non-academic language where possible. Additionally, we have reviewed and proofread our in-text references to ensure accuracy and consistency. Our aim is to produce a high-quality paper that meets the academic standards and expectations of our readers.

---

## [Decision Letter · Decision Letter 1]

7 Aug 2023

PONE-D-23-08938R1The dynamics of carbon emissions, energy, income, and life expectancy: Regional comparative analysisPLOS ONE

Dear Dr. OSEI-KUSI,

Thank you for submitting your manuscript to PLOS ONE. After careful consideration, we feel that it has merit but does not fully meet PLOS ONE’s publication criteria as it currently stands. Therefore, we invite you to submit a revised version of the manuscript that addresses the points raised during the review process.

We look forward to receiving your revised manuscript.

Kind regards,

Hoang Phong Le, Ph.D.

Academic Editor

PLOS ONE

Journal Requirements:

Reviewers' comments:

Reviewer's Responses to Questions

**Comments to the Author**

1. If the authors have adequately addressed your comments raised in a previous round of review and you feel that this manuscript is now acceptable for publication, you may indicate that here to bypass the “Comments to the Author” section, enter your conflict of interest statement in the “Confidential to Editor” section, and submit your "Accept" recommendation.

Reviewer #1: All comments have been addressed

Reviewer #2: (No Response)

Reviewer #3: (No Response)

2. Is the manuscript technically sound, and do the data support the conclusions?

Reviewer #1: Yes

Reviewer #2: (No Response)

Reviewer #3: Partly

3. Has the statistical analysis been performed appropriately and rigorously? 

Reviewer #1: Yes

Reviewer #2: (No Response)

Reviewer #3: Yes

4. Have the authors made all data underlying the findings in their manuscript fully available?

Reviewer #1: Yes

Reviewer #2: (No Response)

Reviewer #3: Yes

5. Is the manuscript presented in an intelligible fashion and written in standard English?

Reviewer #1: Yes

Reviewer #2: (No Response)

Reviewer #3: Yes

6. Review Comments to the Author

Reviewer #1: This article can be published with this form. The authors made all corrections that I mentioned. Congratulations.

Reviewer #2: The author (s) incorporated all the suggested changes. Therefore, I accept this paper in its current form.

Reviewer #3: The authors improve the quality of the paper but the "Introduction, and literature review" section still needs significant changes. The author fails to effectively present the contribution of the study, nor clearly mention the literature gaps of this study from existing literature. Again, policy recommendations need to be redesign alongside the SDGs.

7. PLOS authors have the option to publish the peer review history of their article (what does this mean?). If published, this will include your full peer review and any attached files.

Reviewer #1: No

Reviewer #2: No

Reviewer #3: No

---

## [Author Response · Author response to Decision Letter 1]

18 Sep 2023

Response to reviewers

Reviewer #3: The authors improve the quality of the paper but the "Introduction, and literature review" section still needs significant changes. The author fails to effectively present the contribution of the study, nor clearly mention the literature gaps of this study from existing literature. Again, policy recommendations need to be redesign alongside the SDGs.

Response

We sincerely appreciate your careful review of our paper and your valuable feedback. We are pleased to inform you that we have taken your suggestions into serious consideration and have made substantial revisions to the "Introduction and Literature Review" section of the manuscript.

In response to your concerns, we have worked diligently to enhance the clarity and presentation of the paper's contribution. In the revised section, we have provided a more focused and concise explanation of the study's contribution to the existing literature. We have also taken care to highlight the specific gaps that our study addresses within the context of the existing research landscape. By doing so, we believe the introduction now more effectively communicates the significance of our work.

Furthermore, we have taken your advice to heart regarding the policy recommendations. We have carefully reconsidered and redesigned the policy recommendations section to align more closely with the Sustainable Development Goals (SDGs). This has allowed us to provide a stronger and more actionable set of recommendations that are grounded in the broader framework of sustainable development.

We hope that these revisions address your concerns and greatly enhance the overall quality of the paper. We are genuinely grateful for your constructive feedback, which has undoubtedly played a crucial role in strengthening our manuscript.

---

## [Decision Letter · Decision Letter 2]

13 Oct 2023

The dynamics of carbon emissions, energy, income, and life expectancy: Regional comparative analysis

PONE-D-23-08938R2

Dear Dr. OSEI-KUSI,

We’re pleased to inform you that your manuscript has been judged scientifically suitable for publication and will be formally accepted for publication once it meets all outstanding technical requirements.

Kind regards,

Hoang Phong Le, Ph.D.

Academic Editor

PLOS ONE

Reviewers' comments:

Reviewer's Responses to Questions

**Comments to the Author**

1. If the authors have adequately addressed your comments raised in a previous round of review and you feel that this manuscript is now acceptable for publication, you may indicate that here to bypass the “Comments to the Author” section, enter your conflict of interest statement in the “Confidential to Editor” section, and submit your "Accept" recommendation.

Reviewer #3: All comments have been addressed

2. Is the manuscript technically sound, and do the data support the conclusions?

Reviewer #3: Yes

3. Has the statistical analysis been performed appropriately and rigorously? 

Reviewer #3: Yes

4. Have the authors made all data underlying the findings in their manuscript fully available?

Reviewer #3: Yes

5. Is the manuscript presented in an intelligible fashion and written in standard English?

Reviewer #3: No

6. Review Comments to the Author

Reviewer #3: The authors addressed almost all comments, and it can now be accepted for publication in the PlosOne journal. Thanks

7. PLOS authors have the option to publish the peer review history of their article (what does this mean?). If published, this will include your full peer review and any attached files.

Reviewer #3: No

---

## [Editor Report · Acceptance letter]

8 Nov 2023

PONE-D-23-08938R2 

The dynamics of carbon emissions, energy, income, and life expectancy: regional comparative analysis 

Dear Dr. Osei-Kusi:

I'm pleased to inform you that your manuscript has been deemed suitable for publication in PLOS ONE. Congratulations! Your manuscript is now with our production department. 

Kind regards, 

on behalf of

Dr. Hoang Phong Le 

Academic Editor

PLOS ONE